# The overlapping burden of the three leading causes of disability and death in sub-Saharan African children

Robert C. Reiner Jr. [1,2] ✉, LBD Triple Burden Collaborators* & Simon I. Hay [1,2] ✉

Despite substantial declines since 2000, lower respiratory infections (LRIs), diarrhoeal diseases, and malaria remain among the leading causes of nonfatal and fatal disease burden for children under 5 years of age (under 5), primarily in sub-Saharan Africa (SSA). The spatial burden of each of these diseases has been estimated subnationally across SSA, yet no prior analyses have examined the pattern of their combined burden. Here we synthesise subnational estimates of the burden of LRIs, diarrhoea, and malaria in children under-5 from 2000 to 2017 for 43 sub-Saharan countries. Some units faced a relatively equal burden from each of the three diseases, while others had one or two dominant sources of unit-level burden, with no consistent pattern geographically across the entire subcontinent. Using a subnational counterfactual analysis, we show that nearly 300 million DALYs could have been averted since 2000 by raising all units to their national average. Our findings are directly relevant for decision-makers in determining which and targeting where the most appropriate interventions are for increasing child survival.

More than half of the estimated five million children under the age of 5 (under-5) who died worldwide in 2017 were from sub-Saharan Africa (SSA) (Fig. 1) with substantial between- and within-country variation in these mortality rates across the region[1]. The three most dominant causes of under 5 mortality in SSA in 2017—lower respiratory infections (LRIs), diarrhoeal diseases, and malaria—were responsible for 1,066,000 (95% Uncertainty Interval (UI) 807,000–1,419,000) of these deaths[2], accounting for more than a third (39% (35–44%)) of all child mortality in SSA in that year[3]. In addition to years of life lost (YLLs), sub-Saharan African children were estimated to have a combined nonfatal disease burden of more than 9.6 million (8.1–11.4) years lived with disability (YLDs) in 2017 from these three causes[4]. The total all-cause disease burden for children in SSA from the sum of YLLs and YLDs was 245.6 million (230.7–263.1) disability-adjusted life years (DALYs) in 2017[5], of which over 93.2 million (70.7–124.0; 38%) were due to just these three diseases (Fig. 1). Each of these three dominant causes are largely preventable and treatable and thus represent a potentially avoidable disease burden. The goal of ending preventable child deaths – a primary component of Sustainable Development Goal 3.2[6] – would be supported by identifying and reducing geographical inequality in the distribution of mortality and total disease burden for these three primary drivers at the 2nd administrative level[1] (e.g., 'counties' in Kenya or 'local government areas' in Nigeria, hereinafter referred to as 'units'). This second administrative level is often the implementation unit for intervention planning and delivery.

Previous efforts to map all-cause under-5 mortality patterns have identified areas with persistently high mortality rates[1]. To act on these findings, cause-specific maps of both mortality and total disease burden (DALYs) would be beneficial for all major causes of under-5 mortality to guide specific and/or integrated intervention planning. Further, efforts to co-map individual causes simultaneously can pinpoint locations where total burden remains high despite individual cause reductions, as well as identify which disease is contributing most substantially to the disease burden in any one location. Precision public health efforts of this kind can also serve as an aid in the prioritisation of traditionally vertical public health interventions[7] and could identify opportunities for increasing delivery efficiency. For example, improvements in vaccination require microplanning of subnational

[1]Institute for Health Metrics and Evaluation, University of Washington, Seattle, WA, USA. [2]Department of Health Metrics Sciences, School of Medicine, University of Washington, Seattle, WA, USA. *A list of authors and their affiliations appears at the end of the paper. ✉e-mail: bcreiner@uw.edu; sihay@uw.edu

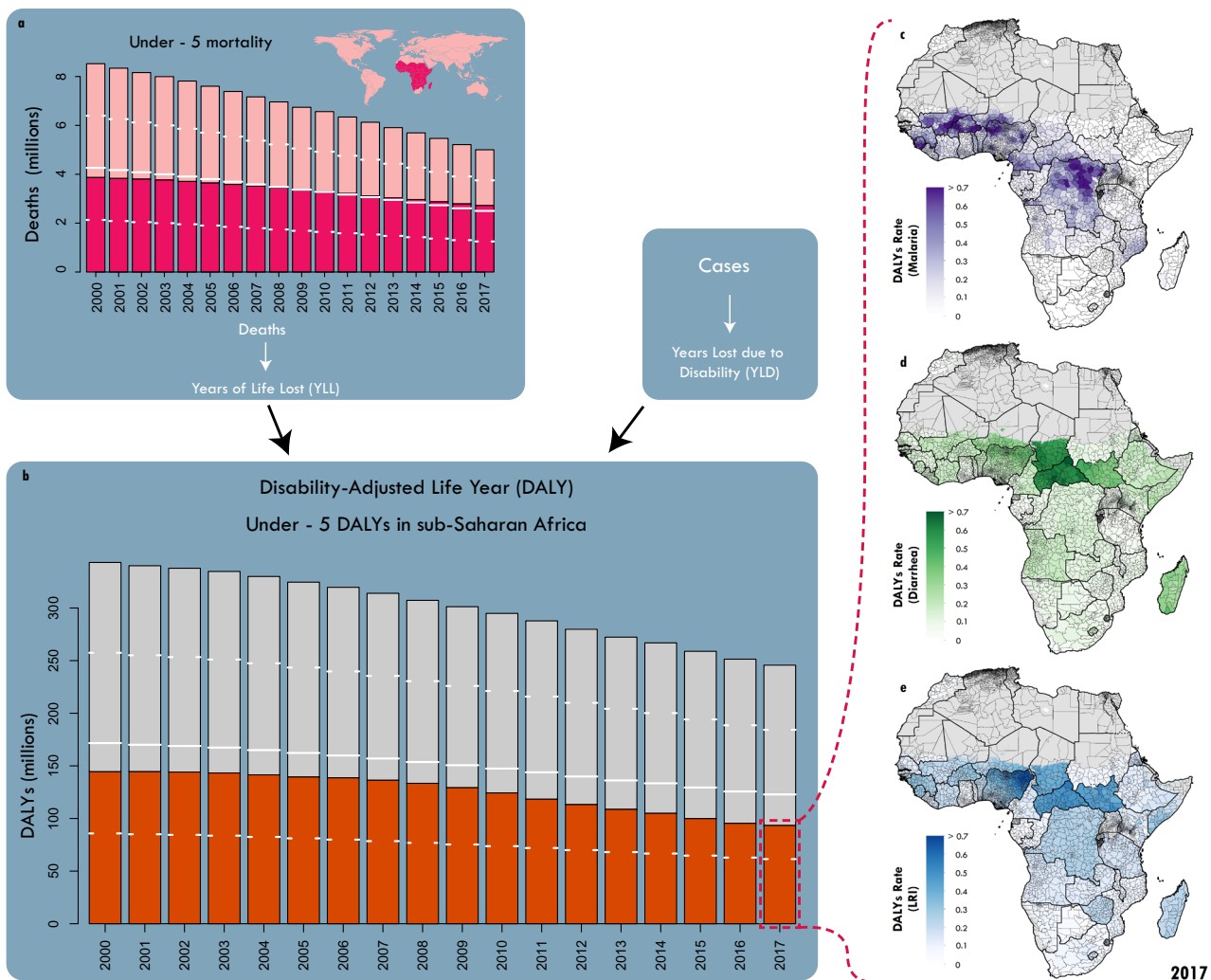

**Fig. 1 | Global and continental trends in under-5 mortality and DALYs. a** Global under-5 mortality from 2000 to 2017. Fuchsia indicates the fraction of those deaths that occur in sub-Saharan Africa. **b** Total under-5 DALYs in sub-Saharan Africa from 2000 to 2017. Orange indicates the fraction of those DALYs attributable to the combined burden of lower respiratory infections (LRIs), diarrhoea, and malaria; Total DALYs per child: estimates of burden attributable to malaria (**c**, purple), diarrhoea (**d**, green), LRIs (**e**, blue), and all other causes (grey) are summed to give total DALY rates for 2017. In all panels, white dashes indicate 25th % and 75th percentiles, and white lines indicate 50%. Maps were produced using ArcGIS Desktop 10.6.

efforts to ensure all children are immunised[8], which in turn relies on knowing which locations are most in need. Using high-resolution estimates of the co-occurring and proportional disease burden from these dominant causes of child burden is thus a key step towards delivering on the promise of precision public health[9,10].

Disease-specific hotspots in death and disability have been previously identified across Africa for each of the three dominant causes of under-5 disease burden separately[11–13]. The analyses illustrate an inconsistent overlap in the geographic units with the highest disease burden from the three causes collectively in 2017 (Fig. 1a-c). Contributing risk factors include a lack of adequate antimalarials and/or insecticide-treated bed nets (ITNs)[14], child growth failure (CGF)[15], micronutrient deficiency[16], poor water quality[17], inadequate sanitation and hygiene[18,19], exposure to household air pollution[20], and poor vaccine coverage[21]. While some risk factors, such as CGF and micronutrient deficiencies, education, and poor access to health systems, have a wide influence across many causes[22], others, such as vaccines or ITNs, are cause-specific. Individual disease maps can be used for targeting intervention campaigns, but they do not help account for the potential efficiencies gained from taking a more systematic approach by considering multiple diseases and their underlying social or health causes simultaneously, which is often a necessity for health programs

in resource-limited settings. Critically, however, hotspots for each disease do not always overlap, and as such, no single suite of interventions across diseases will optimally reduce childhood burden throughout SSA.

Health loss from both nonfatal and fatal disease burden as measured in DALYs is a widely used measure of population health that captures additional health losses beyond mortality[23] (Fig. 1). Given infectious disease DALY burden is predominantly due to years of life lost (YLLs), disease burden estimates will closely reflect deaths due to these diseases. While the burden of each of these three causes has been individually assessed at fine spatial scales in SSA[11,12,24], there remains no comprehensive evaluation of the overlapping burden across the subcontinent or within countries. Here we evaluate patterns in the combined burden of DALYs from LRIs, diarrhoeal diseases, and malaria (hereinafter referred to as 'combined DALYs' or 'combined burden') and assess changes in those patterns over time for children in 43 countries in SSA. We synthesised the most recent estimates, geospatially resolved to the second administrative unit level, of nonfatal and fatal disease burden for each of these three causes[11–13]. We calculated the combined DALYs from LRIs, diarrhoeal diseases, and malaria by summing across causes within each unit. We then conducted a counterfactual analysis comparing median disease-specific burden between

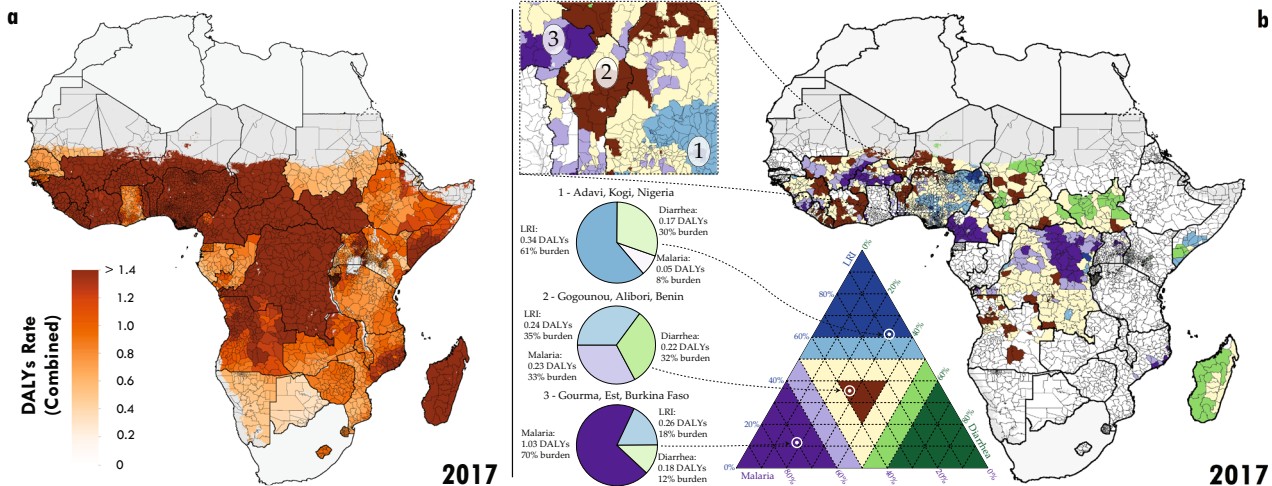

**Fig. 2 | Combined disability-adjusted life years (DALYs) in 2017 and decomposition. a** Second administrative level estimates of combined DALYs. **b** For second administrative units where the combined DALYs per child per year exceeded 0.5 in 2017, the primary component of the local composition of burden is plotted. Units where the combined DALYs were less than 0.5 are plotted as white. Units with dark purple have greater than 60% of their combined burden attributable to malaria (e.g., areas of Burkina Faso). Units with dark blue have greater than 60% of their combined burden attributable to LRIs (e.g., areas of Nigeria). Units with dark green have greater than 60% of their combined burden attributable to diarrhoea (e.g., areas of Chad). Units coloured light purple, blue, or green have between 50% and 60% of their combined burden attributable to malaria, LRIs, or diarrhoea, respectively. Units coloured yellow have no dominant cause (no cause's contribution exceeds 50%). Units which are shaded dark brown have all causes represented in their combined burden with percentages between 20% and 40%. Three examples from Western sub-Saharan Africa are highlighted. For each region, the composition of the relative contribution of each cause translates to a point in the ternary plot legend. The closer a point is to each corner, the higher the relative contribution of that cause. Maps were produced using ArcGIS Desktop 10.6.

subnational units within each country. By using each country's median burden in each year as a local (nationally specific) benchmark, we show for the first time the potential impact of reducing subnational heterogeneity *by improving health outcomes for those most vulnerable* among these three causes on childhood survival. Understanding simultaneous trends in disease burden across these three diseases at the unit level can help local decision-makers direct resources to maximise the impact of health interventions on under-5 mortality and morbidity overall.

## Results

### Patterns of combined burden in 2017

**Patterns of total DALYs.** The average total DALY rate among children across all causes of disease (all-cause DALY rate) in all sub-Saharan African countries in 2017 was 1.5 DALYs per child per year (95% uncertainty interval (UI) 1.5–1.6), accounting for 248.8 million (233.8–266.2) total all-cause DALYs (Fig. 1)[5]. While this burden can be decomposed into infectious and non-communicable diseases as well as injuries, more than half of it was due to infectious diseases (54.7%; 136.0 million (125.1–147.1) DALYs)[5]. Despite having proven and effective interventions[25], LRIs, diarrhoeal diseases, and malaria were associated with about two-thirds of infectious disease burden in SSA (94.1 million (84.1–105.5; 69.2%: (64.2–74.3)) and were responsible for more than a third of the burden among all causes in 2017 (37.8% of all DALYs [34.6–41.4%]). Nigeria, Democratic Republic of the Congo (DRC), and Ethiopia each had more than five million combined DALYs for children due to these three causes, with Nigeria accumulating over 31.4 million (20.6–45.3) combined DALYs in 2017. In terms of combined DALY rate, country-level estimates in Central African Republic (CAR), Chad, Niger, and Sierra Leone all exceeded one DALY per child per year. These countries have the highest DALY rate for children under-5 among all countries globally (the global average DALY rate for LRI, diarrhoea, and malaria was about 0.2 per child-year)[5]. Sixteen countries in SSA had average combined DALY rates above 0.5 per child-year. We used these cut-offs to indicate high (>0.5 DALY per child-year) and very high (>1 DALY per child-year) combined burden.

**Triple DALYs Burden.** As expected, there is a clear pattern where countries with a higher total all-cause burden proportionally had more of that burden attributable to LRIs, diarrhoea, and malaria combined[26] (Supplementaryy Information Fig. 1). Thus, at the country scale, there is a strong suggestion that general interventions that would reduce LRI, diarrhoea, and malaria burden simultaneously would have a massive impact in the countries that have the highest total all-cause childhood disease burden. The distribution of combined DALYs for LRIs, diarrhoea, and malaria exhibits substantial variation between and, importantly, within countries (Fig. 2a). Using mean estimates, eleven countries had at least one unit whose combined mean DALY rate exceed 1 per child per year (Nigeria (297/775 units, 45% of children under-5), CAR (51/51, 100%), Chad (47/55, 89%), DRC (25/213, 6%), Burkina Faso (23/45, 42%), Niger (19/36, 65%), Mali (12/49, 28%), Sierra Leone (9/14, 67%), South Sudan (7/45, 30%), Guinea (2/34, 3%), and Benin (1/64, 2%)). Using the bounds of each country's 95% uncertainty interval on DALYs by unit, we find that as many as 19 and as few as 2 countries may have at least one unit with a combined DALY rate greater than 1 per child. Emphasising the subnational variation, Benin, DRC, Mali, and Nigeria each also had units where the mean estimated combined rate of DALYs was less than 0.5 per child per year. Over half of all sub-Saharan countries included in this analysis (24 of 43) had at least one unit where the mean combined DALY rate was greater than 0.5 per child per year (Fig. 2a), and among these, 19 had subnational variation large enough that some units had at least twice the mean burden of others. The CAR was anomalous in that not only did every unit in the country have an estimated mean combined DALY rate greater than one, but the relative variation (e.g., deviation divided by country average) in the country was low across all countries in SSA.

**Variation in primary cause.** Many of the countries that contained subnational units with high combined burden were the same as those that had high country-level burden for each of the three individual causes. Subnationally, there is no single consistent pattern of which cause or causes contribute the largest share to the combined burden in 2017 (Fig. 2). Among units with high combined burden (rate of over 0.5 DALYs per child per year as presented in colour in Fig. 2b), some were

dominated by a single cause. For example, targeting malaria in the dark purple units in Fig. 2b (such as *Gourma, Est*, Burkina Faso; 66.2% of the combined burden [48.7–80.2%]) or LRIs in the dark blue units in Fig. 2b (such as *Adavi, Kogi*, Nigeria; 52.8% of the combined burden [38.1–66.5%]) might have the largest impact in reducing under-5 DALY burden in those locations. Conversely, there were 13 countries with units with high mean combined burden of LRIs, diarrhoea, and malaria (rate > 0.5 DALYs), yet each cause made an equivalent contribution (between 20% and 40%) to that combined burden. We show these as brown units in Fig. 2b. An example is *Gogounou, Alibori*, Benin (malaria 37.8% [11.6–57.6%], diarrhoea 29.6% [18.0–49.0%], LRIs 32.6% [19.2–46.9%]). These units likely require more universal approaches to burden reduction. While the intensity of combined burden at the second administrative level and at the national level often coincided, such as in CAR, which had the highest country-level mean combined DALY rate (1.4 per child-year [0.9–1.9]) and two of the top 10 highest unit-level combined rates, this was not universal. Locations with high levels of combined LRI, diarrhoea, and malaria burden (Fig. 2a) were not necessarily regionally remarkable in their burden for any single cause (Supplementary Information Tables 1–4). For example, the *Gombe Shani* region of Nigeria had the second-highest combined rate of DALYs in Africa in 2017, but ranked 38th, 294th, and 45th for burden due to LRIs, diarrhoea, and malaria, respectively. Moreover, although Nigeria did not have the highest mean combined DALY rate in Africa in 2017, units within this country had the second-highest and fourth-highest mean combined DALY rates (*Yobe Shani* had 1.78 [0.19–2.9] DALYs per child per year, and *Bayo, Gombe* had 1.76 [0.18–3.0] DALYs per child per year). This exemplifies considerable subnational variation in Nigeria (Supplementary Information Fig. 2). Thus, drawing conclusions on which causes are of most importance (and therefore which interventions are likely optimal) based on national-level statistics can profoundly deviate from conclusions drawn by looking specifically at the subnational units with the highest burden.

Lower respiratory infections were the second leading cause of DALYs globally in SSA among children under-5 in 2017 (after neonatal disorders) and were ubiquitous across SSA. At least 29.6% (18.0–49.0) of the combined DALYs burden in every unit in SSA was attributable to LRIs, which had a population-weighted average contribution of 40.4% (20.3–65.6) of combined DALYs (Supplementary Information Fig. 2). In 2017, LRIs resulted in the greatest total burden of the three causes for children in SSA, but surprisingly they rarely dominated the proportional combined burden in any unit (Fig. 2b). Among the total burden of LRIs, a majority of DALYs occurred in units where LRIs were responsible for just 19.7% (6.9–39.7) of the combined burden occurred (blue regions of Fig. 2b). Only 5.6% (1.6–15.9) of the LRI burden occurred in a unit where LRIs resulted in more than 60% of the combined burden (dark blue regions of Fig. 2b). As such, while LRI is the dominant cause of childhood disease burden in very few units, almost every unit would benefit from including some additional measure of LRI prevention and treatment within their combined intervention strategy.

**Temporal trends 2000 to 2017.** Trends of combined burden also varied substantially between and within countries from 2000 to 2017. In brief, combined LRI, diarrhoea, and malaria disease burden decreased from 2000 to 2017 in almost every unit. Some of the countries with the highest combined burden, such as Niger and Sierra Leone, achieved substantial reductions but via very different pathways (Supplementary Information Fig. 2). Units across Niger reduced both their LRI (56.4% reduction (35.8–71.6)) and diarrhoea burden (64.0% reduction (53.8–72.9)), while their malaria burden did not change significantly over the period (31.2% increase (−23.2–133.6)). Niger introduced the Hib, pneumococcal, and rotavirus childhood vaccines between 2000 and 2017 and had important reductions in childhood growth failure, risk factors that may not have reduced the malaria

disease burden[27,28]. In spite of the uneven trends, the current composition of combined burden across most units of Niger remains mostly equally split across the three causes (33.3% malaria (5.2–49.8), 36.6% diarrhoea (27.7–59.8), 30.2% LRIs (22.4–42.8)) (yellow and brown units in Fig. 2b). Sierra Leone on the other hand, was more consistent in burden reduction across all three causes (27.0% malaria reduction (4.8–44.2), 63.1% diarrhoea reduction (58.0–66.6), 56.1% LRI reduction (44.1–61.2)), possibly due to large reductions in micronutrient (vitamin A and zinc) deficiency[27]. Further, in 2010, Sierra Leone introduced a government program intended to improve access to healthcare (Free Health Care Initiative) which has improved access to and equity for maternal and child health services[29,30].

Although rare, there were instances where disease burden in a unit increased for one of the three causes. There were however a few units where the burden from one of the dominant three diseases increased by so much that it resulted in an overall combined burden increase, most notably Zimbabwe and CAR (Supplementary Information Fig. 3). For units in Zimbabwe, although combined burden is slightly higher in 2017 (0.32 DALYs per child (0.22–0.46)) than 2000 (0.31 DALYs per child (0.21–0.42)), it does appear to have been declining for the past decade (after steep increases in diarrhoea burden between 2000 and 2010, a period of unrest[27] and a cholera epidemic in 2009). Conversely, there are units within CAR where increases have occurred for multiple causes. Since the combined burden in every unit of CAR is high but evenly distributed across all three causes (yellow units in Fig. 2b), national strategy must embrace interventions to reduce all three causes simultaneously. Health conditions in CAR are dire[31]. The country has been embroiled in conflict and civil war for much of the period 2000 to 2017 and the country urgently needs improvements in basic public health measures like childhood nutrition, insecticide-treated bed nets, and childhood vaccines.

### Counterfactual analysis of averted burden

As described above, there is subnational variation in the combined burden, its composition, and its change over time across all countries of sub-Saharan Africa. However, there are clearly some countries with generally higher combined burden and others with generally lower combined burden. Comparing units across countries, to the best-performing unit across the entire continent, would unrealistically assume the ability to achieve reductions far greater than those observed within a country. A more nationally feasible and pragmatic approach is to compare each unit to the median across units within that country and year. Accordingly, for each country for each year, we identified the median LRI burden, diarrhoea burden, and malaria burden and used these as a target benchmark for all other units in that country in that year. Since 2000, there have been an estimated 2.3 billion (2.1–2.5) combined DALYs for children in sub-Saharan Africa due to these three diseases. If every unit in every country had uniformly achieved at least the median burden levels for each disease in its respective country in each year, almost 300 million DALYs (77–743), or about 13% of that burden, could have been averted from 2000 to 2017. In 2017 alone, units performing worse than their country's median burdens for the three diseases combined was responsible for 15.1 million (4.2–38.5) combined DALYs in SSA children (Supplementary Information Table 4). Based on this analysis, several units, particularly in Nigeria and DRC, could have averted a high combined DALY rate in 2017 had they performed at the level of the national median for each of the three diseases in 2017 (Fig. 3a).

More informatively, we see that in countries such as Nigeria, units that could have averted more than 10,000 DALYs over the study period (shown in colour, Fig. 3b, units averting <10,000 DALYs shown in white) would have needed to improve their performance differentially by individual disease. Indeed, the theoretical gains identified by this counterfactual analysis would not be observed uniformly across sub-

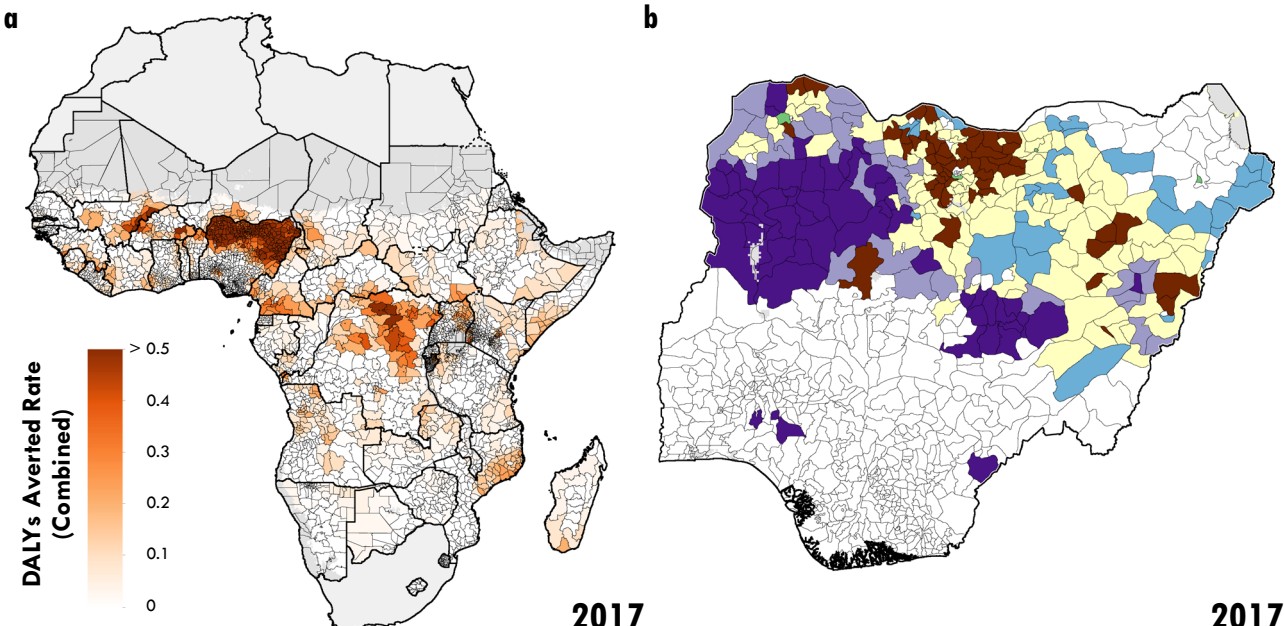

**Fig. 3 | Counterfactual analysis in 2017. a** Second administrative level reductions in combined DALY rates **b** Map of second administrative units in Nigeria whose averted combined DALYs exceeded 10,000, the primary component of the local composition of burden is plotted. Units where the combined DALYs averted were less than 10,000 are plotted as white. Units with dark purple have greater than 60% of their averted combined burden attributable to malaria. Units coloured light purple or blue have between 50% and 60% of their averted combined burden attributable to malaria or LRIs, respectively. Units coloured yellow have no dominant cause (no cause's contribution exceeds 50%). Units which are shaded dark brown have all causes represented in their averted combined burden with percentages between 20% and 40%. Maps were produced using ArcGIS Desktop 10.6.

Saharan Africa, and the sources for these gains vary from country to country and unit to unit. By identifying units where a specific disease has an outsized burden compared to the rest of that country (resulting in a large number of DALYs averted over the study period, shown for example as dark purple units in Fig. 3b for malaria in Nigeria), subnational intervention targeting can be refined to improve equity at the national level. For example, in units that could avert more than 10,000 combined DALYs under the counterfactual, most of the west sub-Saharan and DRC units would achieve this reduction by lowering malaria burden. Conversely, units in Ethiopia and northern Cameroon could experience dramatic declines in DALYs through reductions in DALYs from diarrhoea while north eastern Nigeria would benefit most from reductions in LRIs.

Nigeria – the country with the highest combined burden on the continent and a population that includes 1/6 of all African children – represents a clear example of the impact heterogeneity can have. Of the 31.5 million (20.7–45.3) combined DALYs for Nigerian children in 2017, the elimination of spatial disparity in burden by bringing all units to at least the median rates for each disease would have averted 8.8 million (3.1–17.9; 28%) combined DALYs. Moreover, 308 (140–386) of the 775 units of Nigeria would have experienced a combined burden reduction of more than 10,000 DALYs. Had all units performed at the level of the best-performing unit in each year, in the west of Nigeria, the majority of the averted DALYs would have come from reductions in malaria (Fig. 3b). In parts of the north-east, the reductions would be due to decreasing the high LRI burden while reductions would be due to decreasing the high malaria burden in parts of the northwest and central regions (Fig. 3b.). Crucially, 4.5 million (1.7–8.5) combined DALYs would be averted in units where the reductions are needed across multiple causes (yellow and dark brown in Fig. 3b), with 1.4 million (0.5–2.7) avertable combined DALYs occurring in units with almost identical relative levels of excess across all three causes (dark brown in Fig. 3b). This counterfactual analysis makes clear that interventions that focused on bringing any single cause to its median rate across Nigeria would only address the burden for a subset of the population. Using Nigeria's median performance as the baseline presents a more realistic view of what could be accomplished in a country based on prior experience; crucially, at least some of the wealthier, sub-optimally performing units stand to make improvements.

## Discussion

Three of the most important causes of childhood disability and death in SSA (LRIs, diarrhoea, and malaria)—have substantial variation in the intensity of their combined burden as well as the relative composition of this burden. Our results demonstrate that within-country variation in combined burden manifests differently between countries with high simultaneous burden (as for CAR and Niger), while elsewhere the burden is more concentrated from a single cause (such as diarrhoea in Guinea-Bissau; 58% of combined burden). For many locations, particularly at subnational scales, the combined burden of death and disability from LRIs, diarrhoea, and malaria remains high, even as they reduced cause-specific burden.

There has been substantial progress across SSA in reducing the disease burden from these causes since 2000 but these reductions have not been uniform. While some countries might benefit more than others from targeted interventions that focus on a single cause (such as malaria interventions in Burkina Faso or Sierra Leone), other countries are in need of general interventions across the entirety of the country (such as CAR). Alternately, some countries have a strong need for targeted interventions for one cause in one location, another cause in a second location, as well as general interventions in a third location (such as Nigeria). Understanding the combined burden in any given area can also be valuable in identifying the most effective interventions. For example, LRIs, the largest cause of under-5 burden in SSA among all causes, may be overlooked in these discussions because they are rarely the predominant cause of combined burden at a single location. Local decision-makers could also use these maps to assess whether strategies to deliver focused interventions, such as vaccines or mass drug administration, should be deployed differently in units where they would have the biggest impact (i.e., closing gaps in

coverage or delivering the newly approved RTS,S malaria vaccine which is recommended only for high malaria burden regions[32]).

In 2017 alone, our findings show that more than 13 million (6.9–22.6) combined DALYs occurred in locations where the distribution of DALYs was divided similarly among the three causes. This represents a substantial burden to millions of children in SSA for whom integrated burden-reduction approaches will be required (Fig. 2b). The examples we highlight of areas where decreases in burden over time have occurred for just one of these three dominant sources of death and disability for children under 5, can additionally provide motivation for treatment and prevention efforts that are at once more comprehensive in their scope and more precise in targeting specific locations of greatest need. Considering the overlapping risks related to childhood health (especially those related to LRIs, diarrhoea, and malaria), and the limited availability of resources to address such concerns, interventions that can decrease burden among multiple causes are ideal strategies, effective both in terms of burden reduction and cost[33,34]. Such interventions may span the range of maternal-child health including gender and education equity, maternal micro- and macronutrient fortification, promotion of exclusive breastfeeding, infant zinc and vitamin A supplementation, and complementary feeding programs[25]. Countries with exemplary reductions in under-5 mortality- including Ethiopia, Rwanda, and Senegal- introduced and empowered community or village health workers to connect children with healthcare and improve diagnosis and treatment of infectious diseases[35]. Detailed maps and estimates at subnational scales can show where disease burden is occurring with high precision. As we estimate more causes and risk factors of death and disability at the unit scale, the evidence-base for tailored integrated intervention packages that can reduce burden across multiple causes widens, increasing our capacity to target improvements in childhood health.

Despite the advent of interventions that target U5M directly, such as the mass administration of azithromycin[36], most intervention strategies are planned and delivered vertically. Here we show that as is true at the national level[26], local variation in the major determinants of childhood disability and death will allow greater effectiveness in the geographical targeting of interventions, which can be further catalyzed by a universal platform for intervention delivery. A substantial focus of future research will be investigating how we can scale local analyses for a wider range of the principal determinants of U5M. First, more robust inferential methods must be created in order to identify the optimal interventions at local scales. Combining geo-referenced survey data and an inferential framework, while simultaneously modelling multiple causes in the same overarching scheme, will allow the description of the expected effect of different integrated intervention strategies from an assessment of the covariation of burden. From a methodological standpoint regarding the simultaneous modelling of multiple cases, care must be taken to balance model performance with inferentiality as many causes share the same underlying drivers.

The ability to identify in detail the areas where prevention or intervention strategies have either been successful in reducing combined burden, or conversely reduced a cause-specific burden, can supply valuable case studies and suggest priorities for future interventions. Consistent with our counterfactual analysis, all regions with high burden would certainly benefit from universal improvements in health infrastructure, community case management, and in basic services like sanitation. Almost every country has the potential to improve their performance on at least one of these three diseases in at least some units, based on what has been achieved across all units already. In the absence of needed but costly and sweeping systematic changes in the health landscape across sub-Saharan Africa, the continued development of high-resolution maps of the distribution of disease burden across multiple causes of disease and disability can identify the areas where the greatest improvements in health can be achieved, with the fewest resources. In order to capitalise on these findings most

effectively, local decision-makers could systematically analyse and apply this information in each unit across the African continent. We recognize that matching local policies and infrastructure to our findings would represent a substantial task, but we believe that this work can be used as a guide to focus discussions at the unit level as to which diseases are key to burden reduction. In the future, the availability of a comprehensive set of maps estimating mortality and DALYs for every major cause of under-5 mortality at the unit level, as well as additional efforts to co-map the most substantial contributors to the burden, have the potential to dramatically reduce the remaining preventable under-5 disease burden worldwide.

## Methods

### Disability-adjusted life years (DALYs)
The DALY[37] indicates health loss due to both nonfatal and fatal disease burden, calculated as the sum of years of life lost (YLLs) due to premature mortality and years lived with disability (YLDs). The YLL is based on remaining life expectancy when compared with a reference standard life table at age of death, and the YLD is calculated by multiplying the prevalence of a disease or injury and its main disabling outcomes by its weighted level of severity. As such, one DALY represents one year of healthy life lost. Note: when a location experiences a large amount of under-5 mortality, the average DALYs lost per year can exceed one even though every child who lives can only experience at most one DALY lost per year.

### Estimation of combined DALYs
As discussed above, the individual cause-level spatio-temporally varying estimates of incidence, prevalence, and mortality for LRIs, diarrhoea, and malaria have been previously published[11–13]. Within each paper, the sources, model specifications and model validations are described in detail. Briefly, for malaria, data on incidence, prevalence, and mortality were combined through a number of modelling approaches to form internally consistent estimates. Chief amongst the modelling approaches was a Bayesian geostatistical technique that implemented an integrated nested Laplace approximation (INLA) model on the output of a stacked generalisation product using R-INLA v.20.01.29.9000[38,39]. The INLA model accounted for spatial autocorrelation through the use of a Matérn covariance functional form and for temporal autocorrelation through the use of an autoregressive 1 (AR1) functional form. The INLA model allows both mean and uncertainty estimates to be created in terms of draws from a posterior distribution where individual draws represent a single possible spatiotemporal estimate. For both diarrhoea and LRIs, these estimates were based primarily on survey data assessing prevalence of the associated cause over the two weeks prior to the date of the survey. Again, both stacked generalisation and INLA were sequentially run to arrive at posterior estimates of spatiotemporal burden. For these two causes, a final step was conducted to align prevalence estimates with those of the Global Burden of Disease project[2,5] using logistic raking to ensure internally consistent estimates. The R project v.3.6.1 was used for all analyses. To create combined estimates of DALYs across the three causes, we first estimated YLLs from mortality and YLDs from incidence, both by draw in space-time. As the subnational estimates for each cause are designed to aggregate up to the country-level estimates (using population-weighted aggregation), and the YLLs and YLDs are constant multipliers of deaths and incidence, the mean value of the subnational YLLs and YLDs for each cause and country automatically agree with the corresponding mean country GBD estimate of the respective metric. We then estimated DALYs as above for each individual cause by posterior draw in space-time. As mentioned above, we assumed that the estimates for each cause are independent of one another, and as such to create draw-level estimates of the combined burden we added the draw-level estimates of each cause together. This created combined burden estimates in space and time with

uncertainty. For aggregated estimates, we created population-weighted combined values again at the draw level. Population-based weights were derived from WorldPop estimates of under-5 population[40]. Maps were produced using ArcGIS Desktop 10.6.

## Calculation of annual rate of change (AROC)

For each grid cell, we log-transformed the posterior mean prevalence estimates from each year from 2000 to 2017, $prev^l_{i,yr}$, and determined the rate of change between each pair of adjacent years (beginning with yr=2001):

$$AROC^l_{i,yr} = prev^l_{i,yr} - prev^l_{i,yr-1} \qquad (1)$$

Next, we took a weighted average AROC across the study period, placing more weight on more recent AROCs, and calculated grid-cell-level AROCs. Following Kinyoki et al.[41], weight is defined as:

$$w_{yr} = \frac{(yr - 2000)^\gamma}{\sum_{2001}^{2017}(yr - 2000)^\gamma}, \qquad (2)$$

in which different weights can be given to years across the study period by selecting the appropriate $\gamma$. For this analysis, we chose $\gamma = 1$ for a linear weighting scheme. We calculated grid-cell-level weighted-AROC as:

$$AROC_i = \sum_{2001}^{2017} w_{yr} AROC^l_{i,yr}. \qquad (3)$$

Finally, we calculated unit-level AROC values by taking population-weighted averages of all grid cells in a unit.

## Counterfactual analysis of averted DALYs

We conducted a counterfactual analysis for each country independently. For each year, we identified the subnational unit with the lowest LRI burden, the unit with the lowest diarrhoea burden, and the unit with the lowest malaria burden. Our counterfactual scenario was based on setting every unit of that country at those minimum burden levels. As there are alternative counterfactuals that could be of interest (e.g., comparing each unit to the country average), we have provided the output as a table for others.

## Limitations

Our work, as with any of this scope, comes with a number of limitations. First, we synthesised outputs of three independent modelling exercises into a single large analysis. Within each of those analyses there are limitations that translate to the overall work. In particular, the accuracy of any burden estimation is dependent on the quality and abundance of the data. Data coverage information in each of the three disease-specific models can be found in the supplementary material for each publication (page 26–54[12]; page 45–51[11]; page 37–42[12])[11–13]. This analysis does not address any interactions between the pathogens that cause LRIs, diarrhoea, or malaria. Combining the limitations of data availability and potential overlap in cause-specific burden, we would expect the shared covariates used in the predictive statistical modelling to have similar direction of coefficients (for example, population density) leading to similar disease burden estimates in areas without data coverage. This analysis was intended to be a synthesis of each model, not a joint statistical analysis of each disease burden and so this type of correlation might be expected, especially given some regions within countries have more or fewer resources for public health efforts than others. Because we scaled the subnational disease burden estimates to the national level estimates from the Global Burden of Disease study, we expect there to be some countries with apparent higher or lower magnitude disease burden based on that scalar. Our counterfactual analysis was optimistic in the assumption that the best performing unit within a country is comparable to the rest of the units in the country. One reason for this optimistic approach was a desire to

highlight that, for some locations, their substantial burden is actually average for their country. There are substantial differences between Figs. 3 in terms of observed burden and avertable burden under our counterfactual scenario and these differences specifically highlight the premise that some countries would benefit more from spatially-targeted interventions more than others. A deeper analysis that identifies more realistic strata of risk within a country based on similar disease ecology and then finds the best-performing unit within each strata as the baseline could create fairer comparisons and even more realistic counterfactual estimates. It is also important to note these causes also significantly contribute to DALYs in other age groups. There is a clear need for more detailed data on subnational variation of burden in older age groups as we do not expect the relative overlap of burdens in adults to perfectly match those of children.

## Reporting summary

Further information on research design is available in the Nature Research Reporting Summary linked to this article.

## Data availability

The findings of this study were produced using data available in public online repositories: [https://malariaatlas.org/malaria-burden/], [http://ghdx.healthdata.org/record/ihme-data/africa-under-5-lri-incidence-prevalence-mortality-geospatial-estimates-2000-2017], [http://ghdx.healthdata.org/record/ihme-data/lmic-under-5-diarrhea-incidence-prevalence-and-mortality-geospatial-estimates-2000-2017], [http://ghdx.healthdata.org/record/ihme-data/gbd-2017-incidence-prevalence-and-ylds-1990-2017], data available upon request from the data provider. This study complies with the Guidelines for Accurate and Transparent Health Estimates Reporting (GATHER) recommendations[42]. All maps presented in this study are generated by the authors; no permissions are required for publication.

## Code availability

All code used for the statistical disease burden models is publicly available online at http://ghdx.healthdata.org and on GitHub at https://github.com/ihmeuw/lbd. Code for the counterfactual analysis is available from the Authors upon request.

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

## Acknowledgements

This work was primarily supported by grant OPP1132415 from the Bill & Melinda Gates Foundation.

## Author contributions

Managing the estimation or publication process. L.B.D. T.B.C. Writing the first draft of the manuscript. R.C.R.J. Primary responsibility for this manuscript focused on: applying analytical methods to produce estimates. L.B.D. T.B.C. Primary responsibility for this manuscript focused on: seeking, cataloguing, extracting, or cleaning data; production or coding of figures and tables. L.B.D. T.B.C. Providing data or critical feedback on data sources. L.B.D. T.B.C. and S.I.H. Development of methods or computational machinery. R.C.R.J. and L.B.D. T.B.C. Providing critical feedback on methods or results. L.B.D. T.B.C. and S.I.H. Drafting the manuscript or revising it critically for important intellectual content. R.C.R.J., L.B.D. T.B.C., and S.I.H. Management of the overall research enterprise (for example, through membership in the Scientific Council). L.B.D. T.B.C. and S.I.H. **Consortia author contributions**

**Managing the estimation or publication process**. B.F.B., M.K.M.P. **Writing the first draft of the manuscript**. R.C.R.J. **Primary responsibility for this manuscript focused on: applying analytical methods to produce estimates**. C.A.W. **Primary responsibility for this manuscript focused on: seeking, cataloguing, extracting, or cleaning data; production or coding of figures and tables**. M.M.B. **Providing data or critical feedback on data sources** D.J.W., A.D., C.E.T., H.A., A.E.A., V.A., O.O.A., M.A., B.A., F.A., S.A., H.A., S.M.A., A.A.-H., N.A.-G., A.T.A., S.A., C.L.A., F.A., D.A., S.C.Y.A., J.A., O.A., M.A., F.A., Y.A.A., A.B., M.B., N.B., A.S.B., A.B., V.K.C., D.-T.C., G.D., J.D.G., A.D., S.D.D., M.D., A.E., M.El.S.Z., S.E., T.F., A.M.G., L.G., P.W.G., K.G., A.G., M.G., A.H., S.H., K.H., C.H., H.C.H., M.H., M.H., S.S.N.I., T.J., J.B.J., J.J.J., A.K., G.A.K., Y.S.K., I.A.K., M.N.K., M.K., K.K., M.N.K., Y.J.K., S.K., A.K., N.K., K.K., P.K., D.K., D.K.L., F.H.L., V.C.L., S.L., A.L.-A., K.E.L., S.S.L., P.A.L., X.L., H.M.A.E.R., M.A.M., B.K.M., W.M., R.G.M., E.M.M., B.M., N.M.G.M., S.M., S.M., A.H.M., M.M., A.J.N., J.N., I.N., J.W.N., Q.P.N., H.L.T.N., C.A.N., J.J.N., A.T.O., J.O.O., B.O.O., O.E.O., N.O., S.S.O., M.O.O., M.P.A., J.R.P., A.P., E.K.P., H.Q.P., M.P., M.J.P., H.P., Z.Q.S., F.R., V.R.-M., S.J.R., P.R., S.R., D.L.R., L.R., R.R., A.M.N.R., N.R., J.R., C.M.R.-G., S.S., S.M.S., A.M.S., B.S., D.S., A.A.S., M.A.S., J.I.S., J.A.S., A.S., E.S., C.T.S., S.J.S., D.B.T., A.T.T., B.X.T., P.N.T., B.U., E.U., T.J.V., Y.V., G.T.V., Y.W., R.G.W., T.W., C.S.W., T.G.W., S.Y., Y.G.Y., N.Y., C.Y., H.Y., Z.Z., A.Z., and S.I.H. **Development of methods or computational machinery** R.C.R.J., C.A.W., M.M.B., A.D., L.E., S.B., C.E.T., H.A., D.A., Y.A.A., A.S.B., D.C.C., V.K.C., F.D. A.D., M.D., M.E.S.Z., N.F., J.J.F., P.W.G., M.H., K.B.J., S.K., A., A.D.L., S.M., A.H.M., J.W.N., Q.P.N., S.F.R., A.M.S., E.E.S., S.J.S., E.U., Y.V., K.E.W., Y.G.Y., and N.Y. **Providing critical feedback on methods or results** C.A.W., A.D., C.E.T., H.A., A.E.A., E.A.-G., V.A., O.O.A., M.A., B.A., F.A., Z.A.-A., R.K.A., S.A., H.A., A.A.-H., H.M.A.M., K.A.A., N.A.-Gu., A.T.A., S.A., D.A.A., C.L.A., F.A., D.A., S.C.Y.A., J.A., O.A., M.M.W.A., M.A., F.A., Y.A.A., Z.N.A., A.B., M.B., A.S.B., D.B., N.B., P.B., K.B., O.J.B., Z.A.B., A.B., Z.W.B., A.B., Z.A.B., V.C., M.A.K.C., D.-T.C., C.H.C., G.D., J.D.G., A.H.D., A.D., J.K.D., K.D., A.D., S.D.D., M.D., D.D., S.D., F.D., B.D., L.D.-L., A.E., V.L.F., F.F., N.A.F., M.O.F., M.F., T.F., A.M.G., H.G.G.K.G., L.G., A.A.G., K.E.G., A.G., M.G., F.B.H., S.H., A.H., S.H., C.H., H.C.H., R.H., M.H., M.H., R.H., B.-F.H., S.E.I., O.S.I., I.M.I., M.D.I., S.S.N.I., T.J., R.P.J., J.B.J., J.J.J., A.K., R.K., T.K., A.K., G.A.K., P.N.K., Y.S.K., I.A.K., M.N.K., M.K., K.K., M.M.K., M.N.K., Y.J.K., R.W.K., S.K., A.K., N.K., S.K., A.K., J.A.K., A.K., K.K., P.K., O.P.K., D.K., D.K.L., S.L., K.E.L., S.L., B.L., X.L., A.D.L., H.M.A.E.R., P.W.M., A.A.M., M.A.M., L.B.M., F.R.M.-M., B.K.M., W.M., R.G.M., E.W.M., T.J.M., T.R.M., E.M.M., B.M., N.M.G.M., S.M., S.M., A.H.M., R.M., J.F.M., M.N., A.J.N., J.N., R.N., I.N., J.W.N., H.L.T.N., C.A.N., J.J.N., A.T.O., J.O.O., B.O.O., M.O.O., O.E.O., N.O., S.S.O., M.O.O., M.P.A., J.R.P., A.P., E.K.P., H.Q.P., M.J.P., F.H.P., H.P., Z.Q.S., F.R., V.R.-M., S.J.R., P.R., S.R., D.L.R., L.R., R.R., L.D.R., A.M.N.R., N.R., M.S.R., A.I.R., J.R., C.M.R.-G., S.S., S.M.S., J.A.S., H.S.K., A.M.S., J.S., B.S., D.S., L.E.S., S.S., F.S., A.A.S., M.A.S., A.S., K.S., M.S., J.I.S., B.S., J.A.S., D.L.S., A.S., E.E.S., C.T.S., M.B.S., D.B.T., A.T.T., Y.G.T., M.-H.T., Z.T.T., M.V.T., B.X.T., P.N.T., B.U., E.U., Y.V., F.S.V., G.T.V., Y.W., R.G.W., E.G.W., F.T.W., N.D.W., K.E.W., T.W., C.S.W., T.G.W., T.Y., S.Y., Y.G.Y., P.Y., N.Y., C.Y., D.Y., Z.Z., M.Z., Z.-J.Z., Y.Z., and S.I.H. **Drafting the manuscript or revising it critically for important intellectual content** R.C.R.J., C.A.W., M.K.M.-P., L.E., H.A., E.A.-G., V.A., O.O.A., M.A., B.A. F.A., R.K.A., H.A., A.A.-H., N.A.-G., A.T.A., S.A., D.A.A., R.A., C.L.A., J.A., O.A., M.M.W.A., M.A., F.A., M.A.A., Z.N.A., A.B., A.A.B., M.B., N.B. A.S.B., D.B., K.B., T.T.M.B., O.J.B., J.C., F.C., V.K.C., G.D., A.D., N.D.W., K.D., S.D.D., D.D., E.D., A.E., M.E.S.Z., M.E.T., S.E., V.L.F., E.F., P.F., F.F., N.A.F., M.O.F., M.F., T.F., A.M.G., L.G., A.G., M.I.M.G., D.W.H., A.H., S.H., C.H., H.C.H., R.H., M.H., S.E.I., O.S.I., I.M.I., M.D.I., S.S.N.I., J.J., R.P.J., J.B.J., J.J.J., A.K., A.K., G.A.K., M.N.K., M.K., G.K., K.K., M.M.K., M.N.K., A.K., N.K., A.K., A.K., K.K., P.K., O.P.K., D.K., I.L., S.L., C.L.V., P.H.L., K.E.L., J.L., A.D.L., H.M.A.E.R., P.W.M., A.M., A.A.M., M.A.M., L.B.M., F.R.M.-M., B.G.M., W.M., R.G.M., E.W.M., G.A.M., T.J.M., T.M., T.R.M., B.M., S.M., S.M., A.H.M., R.M., P.M., J.F.M., A.J.N., J.N., I.N., J.W.N., H.L.T.N., V.N.-S., A.T.O., J.O.O., B.O.O., M.O.O., O.E.O., N.O., S.S.O., M.O.O., M.P.A., J.R.P., A.P., H.Q.P., M.J.P., Z.Q.S., F.R., V.R.-M., M.H.U.R., S.J.R., S.R., D.L.R., L.R., N.R., A.I.R., J.R., C.M.R.-G., S.F.R., S.S., J.A.S., H.S.K., A.M.S., J.S., D.S., R.S., M.S., J.A.S., A.S., C.T.S., M.B.S., D.B.T., A.T.T., M.V.T., B.X.T., B.U., E.U., T.J.V., Y.V., F.S.V., G.T.V., R.G.W., N.D.W., K.E.W., T.W., .C.S.W., S.Y., Y.G.Y., Z.Z., M.Z., Z.-J.Z., and S.I.H. **Management of the overall research enterprise (for example, through membership in the Scientific Council)** B.F.B., A.J.C., P.W.G., J.A.K., A.H.M., C.J.L.M., P.C.R., J.A.S., B.S., and S.I.H.

## Competing interests

This study was funded by the Bill & Melinda Gates Foundation. The corresponding author had full access to all the data in the study and had final responsibility for the decision to submit for publication. **The non-consortium authors have no competing interests**. Competing interests for consortium authors is as follows: Robert Ancuceanu reports receiving consultancy or speaker feeds from UCB, Sandoz, Abbvie, Zentiva, Teva, Laropharm, CEGEDIM, Angelini, Biessen Pharma, Hofigal, Astra-Zeneca, and Stada. Jacek Jerzy Jozwiak reports personal fees from Amgen, ALAB Laboratories, Teva, Synexus, Boehringer Ingelheim, and Zentiva, all outside the submitted work. Kewal Krishan reports non-financial support from UGC Centre of Advanced Study, CAS II, Department of Anthropology, Panjab University, Chandigarh, India, outside the submitted work. Walter Mendoza is a Program Analyst in Population and Development at the United Nations Population Fund-UNFPA Country Office in Peru, which does not necessarily endorse or support these findings. Maarten J Postma reports grants and personal fees from MSD, GSK, Pfizer, Boehringer Ingelheim, Novavax, BMS, Seqirus, Astra Zeneca, Sanofi, IQVIA, grants from Bayer, BioMerieux, WHO, EU, FIND, Antilope, DIKTI, LPDP, Budi, personal fees from Novartis, Quintiles, Pharmerit, owning stock options in Health-Ecore and PAG Ltd, and being advisor to Asc Academics, all outside the submitted work. Jasviner A Singh reports personal fees from Crealta/Horizon, Medisys, Fidia, UBM LLC, Trio health, Medscape, WebMD, Clinical Care options, Clearview healthcare partners, Putnam associates, Focus forward, Navigant consulting, Spherix, Practice Point communications, the National Institutes of Health, the American College of Rheumatology, and Simply Speaking, owning stock options in Amarin, Viking, Moderna, Vaxart pharmaceuticals and Charlotte's Web Holdings, being a member of FDA Arthritis Advisory Committee, the steering committee of OMERACT, an international organization that develops measures for clinical trials and receives arm's length funding from 12 pharmaceutical companies, and the Veterans Affairs Rheumatology Field Advisory Committee, and acting as Editor and Director of the UAB Cochrane Musculoskeletal Group Satellite Center on Network Meta-analysis, all outside the submitted work. Era Upadhyay has a patent A system and method of reusable filters for anti-pollution mask pending, and a patent A system and method for electricity generation through crop stubble by using microbial fuel cells pending.

## Additional information

## LBD Triple Burden Collaborators

Robert C. Reiner Jr. [1,2] ✉, Catherine A. Welgan[1], Christopher E. Troeger[1], Mathew M. Baumann[1], Daniel J. Weiss[3], Aniruddha Deshpande[1], Brigette F. Blacker[1], Molly K. Miller-Petrie[1], Lucas Earl[1], Samir Bhatt[4], Hassan Abolhassani[5,6], Akine Eshete Abosetugn[7], Eman Abu-Gharbieh[8], Victor Adekanmbi[9], Olatunji O. Adetokunboh[10,11], Mohammad Aghaali[12], Budi Aji[13], Fares Alahdab[14], Ziyad Al-Aly[15,16], Robert Kaba Alhassan[17], Saqib Ali[18], Hesam Alizade[19], Syed Mohamed Aljunid[20,21], Amir Almasi-Hashiani[22], Hesham M. Al-Mekhlafi[23,24], Khalid A. Altirkawi[25], Nelson Alvis-Guzman[26,27], Azmeraw T. Amare[28,29], Saeed Amini[30], Dickson A. Amugsi[31], Robert Ancuceanu[32], Catalina Liliana Andrei[33], Fereshteh Ansari[34,35], Davood Anvari[36,37], Seth Christopher Yaw Appiah[38,39], Jalal Arabloo[40], Olatunde Aremu[41], Maha Moh'd Wahbi Atout[42], Marcel Ausloos[43,44], Floriane Ausloos[45], Martin Amogre Ayanore[46], Yared Asmare Aynalem[47], Zelalem Nigussie Azene[48], Alaa Badawi[49,50], Atif Amin Baig[51], Maciej Banach[52,53], Neeraj Bedi[54,55], Akshaya Srikanth Bhagavathula[56,57], Dinesh Bhandari[58,59], Nikha Bhardwaj[60], Pankaj Bhardwaj[61,62], Krittika Bhattacharyya[63,64], Zulfiqar A. Bhutta[65,66], Ali Bijani[67], Tesega Tesega Mengistu Birhanu[68], Zebenay Workneh Bitew[69,70], Archith Boloor[71], Oliver J. Brady[72], Zahid A. Butt[73,74], Josip Car[75,76], Felix Carvalho[77], Daniel C. Casey[1], Vijay Kumar Chattu[78,79], Mohiuddin Ahsanul Kabir Chowdhury[80,81], Dinh-Toi Chu[82], Camila H. Coelho[83], Aubrey J. Cook[1], Giovanni Damiani[84,85], Farah Daoud[1], Jiregna Darega Gela[86], Amira Hamed Darwish[87], Ahmad Daryani[88], Jai K. Das[89], Nicole Davis Weaver[1], Kebede Deribe[90,91], Assefa Desalew[92], Samath Dhamminda Dharmaratne[1,2,93], Mostafa Dianatinasab[94,95], Daniel Diaz[96,97], Shirin Djalalinia[98], Fariba Dorostkar[99], Eleonora Dubljanin[100], Bereket Duko[101,102], Laura Dwyer-Lindgren[1,2], Andem Effiong[103], Maysaa El Sayed Zaki[104], Maha El Tantawi[105], Shymaa Enany[106], Nazir Fattahi[107], Valery L. Feigin[1,108,109], Eduarda Fernandes[110], Pietro Ferrara[111], Florian Fischer[112], Nataliya A. Foigt[113], Morenike Oluwatoyin Folayan[114], Masoud Foroutan[115], Joseph Jon Frostad[1], Takeshi Fukumoto[116], Abhay Motiramji Gaidhane[117], Hailemikael Gebrekidan G. K. Gebrekrstos[118], Leake Gebremeskel[119,120], Assefa Ayalew Gebreslassie[121], Peter W. Gething[122,123], Kebede Embaye Gezae[124], Keyghobad Ghadiri[125,126], Ahmad Ghashghaee[40,127], Mahaveer Golechha[128], Mohammed Ibrahim Mohialdeen Gubari[129], Fikaden Berhe Hadgu[130], Samer Hamidi[131], Demelash Woldeyohannes Handiso[132], Abdiwahab Hashi[133], Shoaib Hassan[134,135], Khezar Hayat[136,137], Claudiu Herteliu[44,138], Hung Chak Ho[139], Ramesh Holla[140], Mehdi Hosseinzadeh[141,142], Mowafa Househ[143], Rabia Hussain[144], Bing-Fang Hwang[145], Segun Emmanuel Ibitoye[146], Olayinka Stephen Ilesanmi[147,148], Irena M. Ilic[149], Milena D. Ilic[150], Seyed Sina Naghibi Irvani[151], Jalil Jaafari[152], Tahereh Javaheri[153], Ravi Prakash Jha[154,155], Kimberly B. Johnson[1], Jost B. Jonas[156,157], Jacek Jerzy Jozwiak[158], Ali Kabir[159], Rohollah Kalhor[160,161], Tanuj Kanchan[162], André Karch[163], Gbenga A. Kayode[164,165], Peter Njenga Keiyoro[166], Yousef Saleh Khader[167], Ibrahim A. Khalil[168], Md Nuruzzaman Khan[169], Maseer Khan[170], Gulfaraz Khan[171], Khaled Khatab[172,173], Mona M. Khater[174], Mahalaqua Nazli Khatib[175], Neda Kianipour[176], Yun Jin Kim[177], Ruth W. Kimokoti[178], Sezer Kisa[179], Adnan Kisa[180,181], Niranjan Kissoon[182], Sonali Kochhar[168,183], Ali Koolivand[184], Jacek A. Kopec[185,186], Ai Koyanagi[187,188], Kewal Krishan[189], Pushpendra Kumar[190], Om P. Kurmi[191,192], Dian Kusuma[193,194], Dharmesh Kumar Lal[195], Faris Hasan Lami[196], Iván Landires[197,198], Van Charles Lansingh[199,200], Savita Lasrado[201], Carlo La Vecchia[202], Alice Lazzar-Atwood[1], Paul H. Lee[203], Kate E. LeGrand[1], Sonia Lewycka[204,205], Bingyu Li[206], Stephen S. Lim[1,2], Paulina A. Lindstedt[1], Xuefeng Liu[207], Joshua Longbottom[208], Alan D. Lopez[209], Hassan Magdy Abd El Razek[210], Phetole Walter Mahasha[211], Afshin Maleki[212,213], Abdullah A. Mamun[214], Mohammad Ali Mansournia[215], Laurie B. Marczak[1], Francisco Rogerlândio Martins-Melo[216], Benjamin K. Mayala[1,217], Birhanu Geta Meharie[218], Addisu Melese[219], Walter Mendoza[220], Ritesh G. Menezes[221], Endalkachew Worku Mengesha[222],

George A. Mensah[223,224], Tuomo J. Meretoja[225,226], Tomislav Mestrovic[227,228], Ted R. Miller[102,229], Erkin M. Mirrakhimov[230,231], Babak Moazen[232,233], Naser Mohammad Gholi Mezerji[234], Shadieh Mohammadi[213,235], Shafiu Mohammed[236,232], Ali H. Mokdad[1,2], Masoud Moradi[107], Rahmatollah Moradzadeh[22], Paula Moraga[237], Jonathan F. Mosser[1], Chrisopher J. L. Murray[1,2], Mehdi Naderi[238], Ahamarshan Jayaraman Nagarajan[239,240], Javad Nazari[241], Rawlance Ndejjo[242], Ionut Negoi[243,244], Josephine W. Ngunjiri[245], QuynhAnh P. Nguyen[1], Huong Lan Thi Nguyen[246], Chukwudi A. Nnaji[247,248], Jean Jacques Noubiap[249], Virginia Nuñez-Samudio[250,251], Andrew T. Olagunju[252,253], Jacob Olusegun Olusanya[254], Bolajoko Olubukunola Olusanya[254], Muktar Omer Omer[133], Obinna E. Onwujekwe[255], Nikita Otstavnov[256], Stanislav S. Otstavnov[256,257], Mayowa O. Owolabi[258,259], Mahesh P A[260], Jagadish Rao Padubidri[261], Adrian Pana[44,262], Emmanuel K. Peprah[263], Hai Quang Pham[246], David M. Pigott[1,2], Majid Pirestani[264], Maarten J. Postma[265,266], Faheem Hyder Pottoo[267], Hadi Pourjafar[268,269], Zahiruddin Quazi Syed[117], Fakher Rahim[270,271], Vafa Rahimi-Movaghar[272], Mohammad Hifz Ur Rahman[273], Sowmya J. Rao[274], Puja C. Rao[1], Priya Rathi[140], Salman Rawaf[76,275], David Laith Rawaf[276,277], Lal Rawal[278], Reza Rawassizadeh[279], Lemma Demissie Regassa[280], Andre M. N. Renzaho[281,282], Nima Rezaei[6,283], Mohammad Sadegh Rezai[284], Ana Isabel Ribeiro[285], Jennifer Rickard[286,287], Carlos Miguel Rios-González[288,289], Susan Fred Rumisha[3,290], Siamak Sabour[291], S. Mohammad Sajadi[292,293], Joshua A. Salomon[294], Hossein Samadi Kafil[295], Abdallah M. Samy[296], Juan Sanabria[297,298], Benn Sartorius[2,299], Deepak Saxena[117,300], Lauren E. Schaeffer[301,302], Subramanian Senthilkumaran[303], Feng Sha[304], Amira A. Shaheen[305], Masood Ali Shaikh[306], Rajesh Sharma[307], Aziz Sheikh[308,309], Kenji Shibuya[310], Mika Shigematsu[311], Jae Il Shin[312], Biagio Simonetti[313,314], Jasvinder A. Singh[315,316], David L. Smith[1,2], Amin Soheili[317], Anton Sokhan[318], Emma Elizabeth Spurlock[1], Chandrashekhar T. Sreeramareddy[319], Mu'awiyyah Babale Sufiyan[320], Scott J. Swartz[321,322], Degena Bahrey Tadesse[323], Animut Tagele Tamiru[324], Yonas Getaye Tefera[325], Mohamad-Hani Temsah[25], Zemenu Tadesse Tessema[326], Mariya Vladimirovna Titova[327,328], Bach Xuan Tran[329], Phuong N. Truong[330], Bhaskaran Unnikrishnan[331], Era Upadhyay[332], Tommi Juhani Vasankari[333], Yasser Vasseghian[141], Francesco S. Violante[334,335], Giang Thu Vu[336], Yasir Waheed[337], Richard G. Wamai[338,339], Emebet Gashaw Wassie[340], Fissaha Tekulu Welay[341], Nuwan Darshana Wickramasinghe[342], Kirsten E. Wiens[343], Tissa Wijeratne[344,345], Charles Shey Wiysonge[247,248], Temesgen Gebeyehu Wondmeneh[346], Tomohide Yamada[347], Sanni Yaya[348,349], Yordanos Gizachew Yeshitila[350], Paul Yip[351,352], Naohiro Yonemoto[353,354], Chuanhua Yu[355], Deniz Yuce[356], Hasan Yusefzadeh[357], Zoubida Zaidi[358], Maryam Zamanian[22], Alireza Zangeneh[359], Zhi-Jiang Zhang[360], Yunquan Zhang[361,362], Arash Ziapour[363] & Simon I. Hay [1,2]✉

[3]Malaria Atlas Project, University of Oxford, Oxford, UK. [4]Imperial College London, London, UK. [5]Department of Laboratory Medicine, Karolinska University Hospital, Huddinge, Sweden. [6]Research Center for Immunodeficiencies, Tehran University of Medical Sciences, Tehran, Iran. [7]Department of Public Health, Debre Berhan University, Debre Berhan, Ethiopia. [8]Department of Clinical Sciences, University of Sharjah, Sharjah, United Arab Emirates. [9]Population Health Sciences, King's College London, London, England. [10]Centre of Excellence for Epidemiological Modelling and Analysis, Stellenbosch University, Stellenbosch, South Africa. [11]Department of Global Health, Stellenbosch University, Cape Town, South Africa. [12]Department of Epidemiology and Biostatistics, Qom University of Medical Sciences, Qom, Iran. [13]Faculty of Medicine and Public Health, Jenderal Soedirman University, Purwokerto, Indonesia. [14]Mayo Evidence-based Practice Center, Mayo Clinic Foundation for Medical Education and Research, Rochester, MN, USA. [15]John T. Milliken Department of Internal Medicine, Washington University in St. Louis, St. Louis, MO, USA. [16]Clinical Epidemiology Center, Department of Veterans Affairs, St Louis, MO, USA. [17]Institute of Health Research, University of Health and Allied Sciences, Ho, Ghana. [18]Department of Information Systems, College of Economics and Political Science, Sultan Qaboos University, Muscat, Oman. [19]Infectious and Tropical Disease Research Center, Hormozgan University of Medical Sciences, Bandar Abbas, Iran. [20]Department of Health Policy and Management, Kuwait University, Safat, Kuwait. [21]International Centre for Casemix and Clinical Coding, National University of Malaysia, Bandar Tun Razak, Malaysia. [22]Department of Epidemiology, Arak University of Medical Sciences, Arak, Iran. [23]Medical Research Center, Jazan University, Jazan, Saudi Arabia. [24]Department of Parasitology, Sana'a University, Sana'a, Yemen. [25]Pediatric Intensive Care Unit, King Saud University, Riyadh, Saudi Arabia. [26]Research Group in Health Economics, University of Cartagena, Cartagena, Colombia. [27]Research Group in Hospital Management and Health Policies, ALZAK Foundation, Cartagena, Colombia. [28]School of Medicine, University of Adelaide, Adelaide, SA, Australia. [29]College of Medicine and Health Science, Bahir Dar University, Bahir Dar, Ethiopia. [30]Health Services Management Department, Arak University of Medical Sciences, Arak, Iran. [31]Maternal and Child Wellbeing, African Population and Health Research Center, Nairobi, Kenya. [32]Pharmacy Department, Carol Davila University of Medicine and Pharmacy, Bucharest, Romania. [33]Cardiology Department, Carol Davila University of Medicine and Pharmacy, Bucharest, Romania. [34]Research Center for Evidence Based Medicine, Tabriz University of Medical Sciences, Tabriz, Iran. [35]Razi Vaccine and Serum Research Institute, Agricultural Research, Education, and Extension Organization (AREEO), Tehran, Iran. [36]Department of Parasitology, Mazandaran University of Medical Sciences, Sari, Iran. [37]Department of Parasitology, Iranshahr University of Medical Sciences, Iranshahr, Iran. [38]Department of Sociology and Social Work, Kwame Nkrumah University of Science and Technology, Kumasi, Ghana. [39]Center for International Health, Ludwig Maximilians University, Munich, Germany. [40]Health Management and Economics Research Center, Iran University of Medical Sciences, Tehran, Iran. [41]Department of Public Health, Birmingham City University, Birmingham, UK. [42]Faculty of Nursing, Philadelphia University, Amman, Jordan. [43]School of Business, University of Leicester, Leicester, UK. [44]Department of Statistics and Econometrics, Bucharest University of Economic Studies, Bucharest, Romania. [45]Gastro-enterology Department, University of Liège, Liège, Belgium. [46]Department of Health Policy Planning and Management, University of Health and Allied Sciences, Ho, Ghana. [47]Department of Nursing, Debre Berhan University, Debre Berhan, Ethiopia. [48]Department of Reproductive Health, University of Gondar, Gondar, Ethiopia. [49]Public Health Risk Sciences Division, Public Health Agency of Canada, Toronto, ON, Canada. [50]Department of Nutritional Sciences, University of Toronto, Toronto, ON, Canada. [51]Unit of Biochemistry, Sultan Zainal Abidin University (Universiti Sultan Zainal Abidin), Kuala Terengganu, Malaysia. [52]Department of Hypertension, Medical University of Lodz, Lodz, Poland. [53]Polish Mothers' Memorial Hospital Research Institute, Lodz, Poland. [54]Department of Community Medicine, Gandhi Medical College Bhopal, Bhopal, India.

[55]Jazan University, Jazan, Saudi Arabia. [56]Department of Social and Clinical Pharmacy, Charles University, Hradec Kralova, Czech Republic. [57]Institute of Public Health, United Arab Emirates University, Al Ain, United Arab Emirates. [58]School of Public Health, University of Adelaide, Adelaide, SA, Australia. [59]Public Health Research Laboratory, Tribhuvan University, Kathmandu, Nepal. [60]Department of Anatomy, Government Medical College Pali, Pali, India. [61]Department of Community Medicine and Family Medicine, All India Institute of Medical Sciences, Jodhpur, India. [62]School of Public Health, All India Institute of Medical Sciences, Jodhpur, India. [63]Department of Statistical and Computational Genomics, National Institute of Biomedical Genomics, Kalyani, India. [64]Department of Statistics, University of Calcutta, Kolkata, India. [65]Centre for Global Child Health, University of Toronto, Toronto, ON, Canada. [66]Centre of Excellence in Women & Child Health, Aga Khan University, Karachi, Pakistan. [67]Social Determinants of Health Research Center, Babol University of Medical Sciences, Babol, Iran. [68]Planning, Monitoring and Evaluation Directorate, Amhara Public Health Institute, Bahir Dar, Ethiopia. [69]Nutrition Department, St. Paul's Hospital Millennium Medical College, Addis Ababa, Ethiopia. [70]St. Paul's Hospital Millennium Medical College, Addis Ababa, Ethiopia. [71]Department of Internal Medicine, Manipal Academy of Higher Education, Mangalore, India. [72]Department of Infectious Disease Epidemiology, London School of Hygiene & Tropical Medicine, London, UK. [73]School of Public Health and Health Systems, University of Waterloo, Waterloo, ON, Canada. [74]Al Shifa School of Public Health, Al Shifa Trust Eye Hospital, Rawalpindi, Pakistan. [75]Centre for Population Health Sciences, Nanyang Technological University, Singapore, Singapore. [76]Department of Primary Care and Public Health, Imperial College London, London, UK. [77]Research Unit on Applied Molecular Biosciences (UCIBIO), University of Porto, Porto, Portugal. [78]Department of Medicine, University of Toronto, Toronto, ON, Canada. [79]Global Institute of Public Health (GIPH), Thiruvananthapuram, India. [80]Maternal and Child Health Division, International Centre for Diarrhoeal Disease Research, Bangladesh, Dhaka, Bangladesh. [81]Department of Epidemiology and Biostatistics, University of South Carolina, Columbia, SC, USA. [82]Faculty of Biology, Hanoi National University of Education, Hanoi, Vietnam. [83]Laboratory of Malaria Immunology and Vaccinology, National Institutes of Health, Bethesda, MD, USA. [84]Clinical Dermatology, IRCCS Istituto Ortopedico Galeazzi, University of Milan, Milan, Italy. [85]Department of Dermatology, Case Western Reserve University, Cleveland, OH, USA. [86]Department of Public Health, Ambo University, Ambo, Ethiopia. [87]Department of Pediatrics, Tanta University, Tanta, Egypt. [88]Toxoplasmosis Research Center, Mazandaran University of Medical Sciences, Sari, Iran. [89]Division of Women and Child Health, Aga Khan University, Karachi, Pakistan. [90]Wellcome Trust Brighton and Sussex Centre for Global Health Research, Brighton and Sussex Medical School, Brighton, UK. [91]School of Public Health, Addis Ababa University, Addis Ababa, Ethiopia. [92]School of Nursing and Midwifery, Haramaya University, Harar, Ethiopia. [93]Department of Community Medicine, University of Peradeniya, Peradeniya, Sri Lanka. [94]Department of Epidemiology and Biostatistics, Shahroud University of Medical Sciences, Shahroud, Iran. [95]Department of Epidemiology, Shiraz University of Medical Sciences, Shiraz, Iran. [96]Center of Complexity Sciences, National Autonomous University of Mexico, Mexico City, Mexico. [97]Faculty of Veterinary Medicine and Zootechnics, Autonomous University of Sinaloa, Culiacán Rosales, Mexico. [98]Development of Research and Technology Center, Ministry of Health and Medical Education, Tehran, Iran. [99]Department of Medical Laboratory Sciences, Iran University of Medical Sciences, Tehran, Iran. [100]Institute of Microbiology and Immunology, University of Belgrade, Belgrade, Serbia. [101]School of Public Health, Hawassa University, Hawassa, Ethiopia. [102]School of Public Health, Curtin University, Perth, WA, Australia. [103]Centre Clinical Epidemiology and Biostatistics, University of Newcastle, Newcastle, NSW, Australia. [104]Reference Laboratory of Egyptian Universities Hospitals, Ministry of Higher Education and Research, Cairo, Egypt. [105]Pediatric Dentistry and Dental Public Health Department, Alexandria University, Alexandria, Egypt. [106]Department of Microbiology and Immunology, Suez Canal University, Ismailia, Egypt. [107]Research Center for Environmental Determinants of Health, Kermanshah University of Medical Sciences, Kermanshah, Iran. [108]National Institute for Stroke and Applied Neurosciences, Auckland University of Technology, Auckland, New Zealand. [109]Research Center of Neurology, Moscow, Russia. [110]Associated Laboratory for Green Chemistry (LAQV), University of Porto, Porto, Portugal. [111]Research Center on Public Health, University of Milan Bicocca, Monza, Italy. [112]Institute of Gerontological Health Services and Nursing Research, Ravensburg-Weingarten University of Applied Sciences, Weingarten, Germany. [113]Institute of Gerontology, National Academy of Medical Sciences of Ukraine, Kyiv, Ukraine. [114]Department of Child Dental Health, Obafemi Awolowo University, Ile-Ife, Nigeria. [115]Department of Medical Parasitology, Abadan Faculty of Medical Sciences, Abadan, Iran. [116]Department of Dermatology, Kobe University, Kobe, Japan. [117]Department of Community Medicine, Datta Meghe Institute of Medical Sciences, Wardha, India. [118]Department of Pediatric Nursing, Aksum University, Aksum, Ethiopia. [119]School of Pharmacy, Aksum University, Aksum, Ethiopia. [120]Department of Pharmacy, Mekelle University, Mekelle, Ethiopia. [121]Department of Reproductive Health, Mekelle University, Mekelle, Ethiopia. [122]Telethon Kids Institute, Perth Children's Hospital, Nedlands, WA, Australia. [123]Curtin University, Bentley, WA, Australia. [124]Department of Biostatistics, Mekelle University, Mekelle, Ethiopia. [125]Infectious Disease Research Center, Kermanshah University of Medical Sciences, Kermanshah, Iran. [126]Pediatric Department, Kermanshah University of Medical Sciences, Kermanshah, Iran. [127]Student Research Committee, Iran University of Medical Sciences, Tehran, Iran. [128]Health Systems and Policy Research, Indian Institute of Public Health Gandhinagar, Gandhinagar, India. [129]Department of Family and Community Medicine, University Of Sulaimani, Sulaimani, Iraq. [130]Department of Pediatrics and Child Health, Mekelle University, Mekelle, Ethiopia. [131]School of Health and Environmental Studies, Hamdan Bin Mohammed Smart University, Dubai, United Arab Emirates. [132]Department of Public Health, Wachemo University, Hossana, Ethiopia. [133]Department of Public Health, Jigjiga University, Jijiga, Ethiopia. [134]Center for International Health (CIH), University of Bergen, Bergen, Norway. [135]Bergen Center for Ethics and Priority Setting (BCEPS), University of Bergen, Bergen, Norway. [136]Institute of Pharmaceutical Sciences, University of Veterinary and Animal Sciences, Lahore, Pakistan. [137]Department of Pharmacy Administration and Clinical Pharmacy, Xian Jiaotong University, Xian, China. [138]School of Business, London South Bank University, London, UK. [139]Department of Urban Planning and Design, University of Hong Kong, Hong Kong, China. [140]Kasturba Medical College, Mangalore, Manipal Academy of Higher Education, Manipal, India. [141]Institute of Research and Development, Duy Tan University, Da Nang, Vietnam. [142]Department of Computer Science, University of Human Development, Sulaymaniyah, Iraq. [143]College of Science and Engineering, Hamad Bin Khalifa University, Doha, Qatar. [144]School of Pharmaceutical Sciences, University of Science Malaysia, Penang, Malaysia. [145]Department of Occupational Safety and Health, China Medical University, Taichung, Taiwan. [146]Department of Health Promotion and Education, University of Ibadan, Ibadan, Nigeria. [147]Department of Community Medicine, University of Ibadan, Ibadan, Nigeria. [148]Department of Community Medicine, University College Hospital, Ibadan, Ibadan, Nigeria. [149]Faculty of Medicine, University of Belgrade, Belgrade, Serbia. [150]Department of Epidemiology, University of Kragujevac, Kragujevac, Serbia. [151]Research Institute for Endocrine Sciences, Shahid Beheshti University of Medical Sciences, Tehran, Iran. [152]Department of Environmental Health Engineering, Guilan University of Medical Sciences, Rasht, Iran. [153]Health Informatic Lab, Boston University, Boston, MA, USA. [154]Department of Community Medicine, Dr. Baba Saheb Ambedkar Medical College & Hospital, Delhi, India. [155]Department of Community Medicine, Banaras Hindu University, Varanasi, India. [156]Department of Ophthalmology, Heidelberg University, Heidelberg, Germany. [157]Beijing Institute of Ophthalmology, Beijing Tongren Hospital, Beijing, China. [158]Department of Family Medicine and Public Health, University of Opole, Opole, Poland. [159]Minimally Invasive Surgery Research Center, Iran University of Medical Sciences, Tehran, Iran. [160]Institute for Prevention of Non-communicable Diseases, Qazvin University of Medical Sciences, Qazvin, Iran. [161]Health Services Management Department, Qazvin University of Medical Sciences, Qazvin, Iran. [162]Department of Forensic Medicine and Toxicology, All India Institute of Medical Sciences, Jodhpur, India. [163]Institute for Epidemiology and Social Medicine, University of Münster, Münster, Germany. [164]International Research Center of Excellence, Institute of Human Virology Nigeria, Abuja, Nigeria. [165]Julius Centre for Health Sciences and Primary Care, Utrecht University, Utrecht, Netherlands. [166]Open, Distance and eLearning Campus, University of Nairobi, Nairobi, Kenya. [167]Department of

Public Health, Jordan University of Science and Technology, Irbid, Jordan. [168]Department of Global Health, University of Washington, Seattle, WA, USA. [169]Department of Population Science, Jatiya Kabi Kazi Nazrul Islam University, Mymensingh, Bangladesh. [170]Epidemiology Department, Jazan University, Jazan, Saudi Arabia. [171]Department of Medical Microbiology & Immunology, United Arab Emirates University, Al Ain, United Arab Emirates. [172]Faculty of Health and Wellbeing, Sheffield Hallam University, Sheffield, UK. [173]College of Arts and Sciences, Ohio University, Zanesville, OH, USA. [174]Department of Medical Parasitology, Cairo University, Cairo, Egypt. [175]Global Evidence Synthesis Initiative, Datta Meghe Institute of Medical Sciences, Wardha, India. [176]Department of Public Health, Kermanshah University of Medical Sciences, Kermanshah, Iran. [177]School of Traditional Chinese Medicine, Xiamen University Malaysia, Sepang, Malaysia. [178]Department of Nutrition, Simmons University, Boston, MA, USA. [179]Department of Nursing and Health Promotion, Oslo Metropolitan University, Oslo, Norway. [180]School of Health Sciences, Kristiania University College, Oslo, Norway. [181]Global Community Health and Behavioral Sciences, Tulane University, New Orleans, LA, USA. [182]Department of Pediatrics, University of British Columbia, Vancouver, BC, Canada. [183]Global Healthcare Consulting, New Delhi, India. [184]Department of Environmental Health Engineering, Arak University of Medical Sciences, Arak, Iran. [185]School of Population and Public Health, University of British Columbia, Vancouver, BC, Canada. [186]Arthritis Research Canada, Richmond, BC, Canada. [187]CIBERSAM, San Juan de Dios Sanitary Park, Sant Boi de Llobregat, Spain. [188]Catalan Institution for Research and Advanced Studies (ICREA), Barcelona, Spain. [189]Department of Anthropology, Panjab University, Chandigarh, India. [190]International Institute for Population Sciences, Mumbai, India. [191]Faculty of Health and Life Sciences, Coventry University, Coventry, UK. [192]Department of Medicine, McMaster University, Hamilton, ON, Canada. [193]Imperial College Business School, Imperial College London, London, UK. [194]Faculty of Public Health, University of Indonesia, Depok, Indonesia. [195]Public Health Foundation of India, Gurugram, India. [196]Department of Community and Family Medicine, University of Baghdad, Baghdad, Iraq. [197]Unit of Genetics and Public Health, Institute of Medical Sciences, Las Tablas, Panama. [198]Ministry of Health, Herrera, Panama. [199]Medical Director, HelpMeSee, New York, NY, USA. [200]General Director, Mexican Institute of Ophthalmology, Queretaro, Mexico. [201]Department of Otorhinolaryngology, Father Muller Medical College, Mangalore, India. [202]Department of Clinical Sciences and Community Health, University of Milan, Milan, Italy. [203]School of Nursing, Hong Kong Polytechnic University, Hong Kong, China. [204]Centre for Tropical Medicine and Global Health, University of Oxford, Oxford, UK. [205]Oxford University Clinical Research Unit, Wellcome Trust Asia Programme, Hanoi, Vietnam. [206]Department of Sociology, Shenzhen University, Shenzhen, China. [207]Department of Systems, Populations, and Leadership, University of Michigan, Ann Arbor, MI, USA. [208]Department of Vector Biology, Liverpool School of Tropical Medicine, Liverpool, UK. [209]Independent Consultant, Melbourne, VIC, Australia. [210]Radiology Department, Egypt Ministry of Health and Population, Mansoura, Egypt. [211]Grants, Innovation and Product Development Unit, South African Medical Research Council, Cape Town, South Africa. [212]Environmental Health, Tehran University of Medical Sciences, Tehran, Iran. [213]Environmental Health Research Center, Kurdistan University of Medical Sciences, Sanandaj, Iran. [214]Institute for Social Science Research, The University of Queensland, Indooroopilly, QLD, Australia. [215]Department of Epidemiology and Biostatistics, Tehran University of Medical Sciences, Tehran, Iran. [216]Campus Caucaia, Federal Institute of Education, Science and Technology of Ceará, Caucaia, Brazil. [217]ICF International, DHS Program, Rockville, MD, USA. [218]Department of Pharmacy, Wollo University, Dessie, Ethiopia. [219]Department of Medical Laboratory Sciences, Bahir Dar University, Bahir Dar, Ethiopia. [220]Peru Country Office, United Nations Population Fund (UNFPA), Lima, Peru. [221]Forensic Medicine Division, Imam Abdulrahman Bin Faisal University, Dammam, Saudi Arabia. [222]Department of Reproductive Health and Population Studies, Bahir Dar University, Bahir Dar, Ethiopia. [223]Center for Translation Research and Implementation Science, National Institutes of Health, Bethesda, MD, USA. [224]Department of Medicine, University of Cape Town, Cape Town, South Africa. [225]Breast Surgery Unit, Helsinki University Hospital, Helsinki, Finland. [226]University of Helsinki, Helsinki, Finland. [227]Clinical Microbiology and Parasitology Unit, Dr. Zora Profozic Polyclinic, Zagreb, Croatia. [228]University Centre Varazdin, University North, Varazdin, Croatia. [229]Pacific Institute for Research & Evaluation, Calverton, MD, USA. [230]Internal Medicine Programme, Kyrgyz State Medical Academy, Bishkek, Kyrgyzstan. [231]Department of Atherosclerosis and Coronary Heart Disease, National Center of Cardiology and Internal Disease, Bishkek, Kyrgyzstan. [232]Heidelberg Institute of Global Health (HIGH), Heidelberg University, Heidelberg, Germany. [233]Institute of Addiction Research (ISFF), Frankfurt University of Applied Sciences, Frankfurt, Germany. [234]Department of Biostatistics, Hamadan University of Medical Sciences, Hamadan, Iran. [235]Research Institute for Health Development, Kurdistan University of Medical Sciences, Sanandaj City, Iran. [236]Health Systems and Policy Research Unit, Ahmadu Bello University, Zaria, Nigeria. [237]Computer, Electrical, and Mathematical Sciences and Engineering Division, King Abdullah University of Science and Technology, Thuwal, Saudi Arabia. [238]Clinical Research Development Center, Kermanshah University of Medical Sciences, Kermanshah, Iran. [239]Research and Analytics Department, Initiative for Financing Health and Human Development, Chennai, India. [240]Department of Research and Analytics, Bioinsilico Technologies, Chennai, India. [241]Department of Pediatrics, Arak University of Medical Sciences, Arak, Iran. [242]Disease Control and Environmental Health, Makerere University, Kampala, Uganda. [243]Department of General Surgery, Carol Davila University of Medicine and Pharmacy, Bucharest, Romania. [244]Department of General Surgery, Emergency Hospital of Bucharest, Bucharest, Romania. [245]Department of Biological Sciences, University of Embu, Embu, Kenya. [246]Institute for Global Health Innovations, Duy Tan University, Hanoi, Vietnam. [247]South African Medical Research Council, Cape Town, South Africa. [248]School of Public Health and Family Medicine, University of Cape Town, Cape Town, South Africa. [249]Centre for Heart Rhythm Disorders, University of Adelaide, Adelaide, SA, Australia. [250]Unit of Microbiology and Public Health, Institute of Medical Sciences, Las Tablas, Panama. [251]Department of Public Health, Ministry of Health, Herrera, Panama. [252]Department of Psychiatry and Behavioural Neurosciences, McMaster University, Hamilton, ON, Canada. [253]Department of Psychiatry, University of Lagos, Lagos, Nigeria. [254]Centre for Healthy Start Initiative, Lagos, Nigeria. [255]Department of Pharmacology and Therapeutics, University of Nigeria Nsukka, Enugu, Nigeria. [256]Laboratory of Public Health Indicators Analysis and Health Digitalization, Moscow Institute of Physics and Technology, Dolgoprudny, Russia. [257]Department of Project Management, National Research University Higher School of Economics, Moscow, Russia. [258]Department of Medicine, University of Ibadan, Ibadan, Nigeria. [259]Department of Medicine, University College Hospital, Ibadan, Ibadan, Nigeria. [260]Department of Respiratory Medicine, Jagadguru Sri Shivarathreeswara Academy of Health Education and Research, Mysore, India. [261]Department of Forensic Medicine, Manipal Academy of Higher Education, Mangalore, India. [262]Department of Health Metrics, Center for Health Outcomes & Evaluation, Bucharest, Romania. [263]School of Global Public Health, New York University, New York, NY, USA. [264]Department of Parasitology and Entomology, Tarbiat Modares University, Tehran, Iran. [265]University Medical Center Groningen, University of Groningen, Groningen, Netherlands. [266]School of Economics and Business, University of Groningen, Groningen, Netherlands. [267]Department of Pharmacology, Imam Abdulrahman Bin Faisal University, Dammam, Saudi Arabia. [268]Department of Nutrition and Food Sciences, Maragheh University of Medical Sciences, Maragheh, Iran. [269]Dietary Supplements and Probiotic Research Center, Alborz University of Medical Sciences, Karaj, Iran. [270]Thalassemia and Hemoglobinopathy Research Center, Ahvaz Jundishapur University of Medical Sciences, Ahvaz, Iran. [271]Metabolomics and Genomics Research Center, Tehran University of Medical Sciences, Tehran, Iran. [272]Sina Trauma and Surgery Research Center, Tehran University of Medical Sciences, Tehran, Iran. [273]Department of Community Medicine, Maharishi Markandeshwar Medical College & Hospital, Solan, India. [274]Department of Oral Pathology, Srinivas Institute of Dental Sciences, Mangalore, India. [275]Academic Public Health England, Public Health England, London, UK. [276]WHO Collaborating Centre for Public Health Education and Training, Imperial College London, London, UK. [277]University College London Hospitals, London, UK. [278]School of Health, Medical and Applied Sciences, CQ University, Sydney, NSW, Australia. [279]Department of Computer Science, Boston University, Boston, MA, USA. [280]School of Public Health, Haramaya University, Harar, Ethiopia. [281]School of Social Sciences and Psychology, Western Sydney University, Penrith, NSW, Australia.

[282]Translational Health Research Institute, Western Sydney University, Penrith, NSW, Australia. [283]Network of Immunity in Infection, Malignancy and Auto-immunity (NIIMA), Universal Scientific Education and Research Network (USERN), Tehran, Iran. [284]Pediatric Infectious Diseases Research Center, Mazandaran University of Medical Sciences, Sari, Iran. [285]Epidemiology Research Unit Institute of Public Health (EPIUnit-ISPUP), University of Porto, Porto, Portugal. [286]Department of Surgery, University of Minnesota, Minneapolis, MN, USA. [287]Department of Surgery, University Teaching Hospital of Kigali, Kigali, Rwanda. [288]Faculty of Medical Sciences, Research Department, National University of Caaguazu, Cnel. Oviedo, Paraguay. [289]Department of Research and Publications, National Institute of Health, Asunción, Paraguay. [290]Department of Health Statistics, National Institute for Medical Research, Dar es Salaam, Tanzania. [291]Department of Epidemiology, Shahid Beheshti University of Medical Sciences, Tehran, Iran. [292]Department of Phytochemistry, Soran University, Soran, Iraq. [293]Department of Nutrition, Cihan University-Erbil, Kurdistan Region, Iraq. [294]Center for Health Policy & Center for Primary Care and Outcomes Research, Stanford University, Stanford, CA, USA. [295]Drug Applied Research Center, Tabriz University of Medical Sciences, Tabriz, Iran. [296]Department of Entomology, Ain Shams University, Cairo, Egypt. [297]Department of Surgery, Marshall University, Huntington, WV, USA. [298]Department of Nutrition and Preventive Medicine, Case Western Reserve University, Cleveland, OH, USA. [299]Faculty of Infectious and Tropical Diseases, London School of Hygiene & Tropical Medicine, London, UK. [300]Department of Epidemiology, Indian Institute of Public Health, Gandhinagar, India. [301]Global Programs, Medical Teams International, Seattle, WA, USA. [302]Department of Pediatric Newborn Medicine, Brigham and Women's Hospital, Boston, MA, USA. [303]Emergency Department, Manian Medical Centre, Erode, India. [304]Center for Biomedical Information Technology, Shenzhen Institutes of Advanced Technology, Shenzhen, China. [305]Public Health Division, An-Najah National University, Nablus, Palestine. [306]Independent Consultant, Karachi, Pakistan. [307]University School of Management and Entre-preneurship, Delhi Technological University, Delhi, India. [308]Centre for Medical Informatics, University of Edinburgh, Edinburgh, UK. [309]Division of General Internal Medicine, Harvard University, Boston, MA, USA. [310]Institute for Population Health, King's College London, London, UK. [311]National Institute of Infectious Diseases, Tokyo, Japan. [312]College of Medicine, Yonsei University, Seoul, South Korea. [313]Department of Law, Economics, Management and Quantitative Methods, University of Sannio, Benevento, Italy. [314]WSB University in Gdańsk, Gdansk, Poland. [315]School of Medicine, University of Alabama at Birmingham, Birmingham, AL, USA. [316]Medicine Service, US Department of Veterans Affairs (VA), Birmingham, AL, USA. [317]Nursing Care Research Center, Semnan University of Medical Sciences, Semnan, Iran. [318]Department of Infectious Diseases, Kharkiv National Medical University, Kharkiv, Ukraine. [319]Division of Community Medicine, International Medical University, Kuala Lumpur, Malaysia. [320]Department of Community Medicine, Ahmadu Bello University, Zaria, Nigeria. [321]School of Medicine, University of California San Francisco, San Francisco, CA, USA. [322]Joint Medical Program, University of California Berkeley, Berkeley, CA, USA. [323]Department of Nursing, Aksum University, Aksum, Ethiopia. [324]Department of Midwifery, University of Gondar, Gondar, Ethiopia. [325]Department of Clinical Pharmacy, University of Gondar, Gondar, Ethiopia. [326]Department of Epidemiology and Biostatistics, University of Gondar, Gondar, Ethiopia. [327]K.A. Timiryazev Institute of Plant Physiology, Russian Academy of Sciences, Moscow, Russia. [328]Laboratory of Public Health Indicators Analysis and Health Digitalization, Moscow Institute of Physics and Technology, Moscow, Russia. [329]Department of Health Economics, Hanoi Medical University, Hanoi, Vietnam. [330]Faculty of Geo-Information Science and Earth Observation, University of Twente, Enschede, Netherlands. [331]Kasturba Medical College, Manipal Academy of Higher Education, Mangalore, India. [332]Amity Institute of Biotechnology, Amity University Rajasthan, Jaipur, India. [333]UKK Institute, Tampere, Finland. [334]Department of Medical and Surgical Sciences, University of Bologna, Bologna, Italy. [335]Occupational Health Unit, Sant'Orsola Malpighi Hospital, Bologna, Italy. [336]Center of Excellence in Behavioral Medicine, Nguyen Tat Thanh University, Ho Chi Minh City, Vietnam. [337]Foundation University Medical College, Foundation University Islamabad, Islamabad, Pakistan. [338]Cultures, Societies and Global Studies, & Integrated Initiative for Global Health, Northeastern University, Boston, MA, USA. [339]School of Public Health, University of Nairobi, Nairobi, Kenya. [340]Department of Human Nutrition and Food Sciences, Debre Markos University, Debre Markos, Ethiopia. [341]Department of Midwifery, Adigrat University, Adigrat, Ethiopia. [342]Department of Community Medicine, Rajarata University of Sri Lanka, Anuradhapura, Sri Lanka. [343]Department of Epidemiology, Johns Hopkins University, Baltimore, MD, USA. [344]Department of Neurology, University of Melbourne, Melbourne, VIC, Australia. [345]Department of Medicine, University of Rajarata, Saliyapura Anuradhapuraya, Sri Lanka. [346]Department of Public Health, Samara University, Samara, Ethiopia. [347]Department of Diabetes and Metabolic Diseases, University of Tokyo, Tokyo, Japan. [348]School of International Development and Global Studies, University of Ottawa, Ottawa, ON, Canada. [349]The George Institute for Global Health, University of Oxford, Oxford, UK. [350]Department of Nursing, Arba Minch University, Arba Minch, Ethiopia. [351]Centre for Suicide Research and Prevention, University of Hong Kong, Hong Kong, China. [352]Department of Social Work and Social Administration, University of Hong Kong, Hong Kong, China. [353]Department of Neuropsychopharmacology, National Center of Neurology and Psychiatry, Kodaira, Japan. [354]Department of Public Health, Juntendo University, Tokyo, Japan. [355]Department of Epidemiology and Biostatistics, Wuhan University, Wuhan, China. [356]Cancer Institute, Hacettepe University, Ankara, Turkey. [357]Department of Health Care Management and Economics, Urmia University of Medical Science, Urmia, Iran. [358]Department of Medicine, University Ferhat Abbas of Setif, Sétif, Algeria. [359]Social Development and Health Promotion Research Center, Kermanshah University of Medical Sciences, Kermanshah, Iran. [360]School of Medicine, Wuhan University, Wuhan, China. [361]School of Public Health, Wuhan University of Science and Technology, Wuhan, China. [362]Hubei Province Key Laboratory of Occupational Hazard Identification and Control, Wuhan University of Science and Technology, Wuhan, China. [363]Department of Health Education and Health Promotion, Kermanshah University of Medical Sciences, Kermanshah, Iran.

