## [Peer Review File · Nature Communications]

The overlapping burden of the three leading causes of disability and death in sub-Saharan African childrenEditorial Note: This manuscript has been previously reviewed at another journal that is not operating a transparent peer review scheme. This document only contains reviewer comments and rebuttal letters for versions considered at *Nature Communications*.

REVIEWER COMMENTS

Reviewer #3 (Remarks to the Author):

I believe the paper requires a major rewrite in terms of both the scientific finding and analysis methods, in order for it to be suitable for publication. I thank the authors to respond to my previous comments. However, I find the response to be usually not sufficient. I do agree the efforts made by the authors are admirable, and this paper may have an audience somewhere, but the goals of the paper is beyond what analysis and evidence in this paper could possibly support.

1. In the response, the authors argue the primary aim for the paper "was to highlight these sorts of differences (the different split of burdens spatially)" and the "counterfactual analysis is one approach to highlight the resulting estimates". However, I do think to most readers, the current form of the paper places very strong emphasis on the counterfactual analysis (and it is the only 'new analysis' as well). This is troubling to me as while the authors acknowledge the counterfactual target is 'optimistic', I would argue they are 'misleading'. Finding the best Admin-2 area in a country is a lazy target that means different things in different country: whether it is achievable goal? Are you basing the counterfactual scenario on a capitol urban administrative unit that is largely different from the entire rest of the country? Or are you basing the scenario on a lowly populated unit with highly uncertain estimates? The point here, is not whether the target is optimistic, but rather the choice of the counterfactual is heterogeneous and uninterpretable when comparing across countries. So by presenting the approach and estimates for sub-Saharan Africa as a whole, I think it is misleading to the audience.

2. While the previous comment might be addressed by country-specific analysis, which is not the purpose of this paper, but could be done if someone put in the effort. A bigger concern I have is the validity of the general methodology. The authors in the response argued that the multiple spatial surface are substantially different, and thus are suitable for the analysis where they treat the three sets of estimates as independent. I am not convinced the authors fully understand my concern. Dependence and 'how similar the posterior means are on the map' measures two different things. Any covariates and input shared in the three models would lead to dependences in the estimates, as any uncertainties in those covariates would propagate through to all the final estimates. If there are a large number of overlapping covariates used, the estimates will be highly correlated. In addition, the fact that spatial and temporal smoothing models are used also lead to common unmeasured confounders being captured by the random effects in the three models. Both could be very difficult to quantify after fitting three models separately. Thus it seems to me that treating the three analysis as independent could potentially both mess up the uncertainty of the final joint analysis, and lead to bias in the conclusion. Finally, while as I emphasized already, looking at Figure 1 cannot prove/illustrate independence, I want to highlight the similarities of the Diarrhea and LRI map, which I would not think as 'quite different', and certainly would consider the chance of it being driven by modeling (in addition to the true co-occurrence) to be non-ignorable.

Reviewer #4 (Remarks to the Author):

This research calculates DALYs for 3 or the major causes of under 5 morbidity and mortality across 43 sub-Saharan African countries to the second administrative unit to understand their co-occurrence. The finding that there is a substantial amount of heterogeneity in disease patterns across and between countries supports the authors conclusions of the need for a more tailored approach to maximise the potential for impact in reducing disease burden. This work will be influential in the growing recognition of the potential for integrated disease surveillance and control. The authors have adequately responded to the comments from the previous review and I only have a couple of additional comments for consideration.

- The disease-specific models applied for this analysis have been previously published and methods used have been adequately summarised here. However, the models are ultimately reliant on the availability and quality of data on each of the three diseases being assessed. Although the issue of data quality is briefly acknowledged in the limitations paragraph, it would be helpful to present the results in a way where the data quality issues over time and between countries are better acknowledged.
- Shifting the focus from vertical programs to a more integrated and data-driven perspective on reducing disease-burden is an important finding. However, for programs to inform interventions tailored to the distribution of multiple diseases, the data that is typically available is case/death counts per unit and not DALYs or modelled estimates that can smooth over data quality issues or combining active and passive surveillance data as per the malaria analysis presented. How could the observed trends translate into a more operational approach accessible to programs? This is touched upon in lines 300-304 but could be expanded from a more operational perspective (e.g. could programs simply map each disease counts independently and compare the patterns?).
- the decision for selecting the benchmark for comparison and the limitations are well explained (as per the previous reviewers comments). However, I think one point that doesn't come out very well is the heterogeneity of disease burden in the country impacting how easy it is to achieve the national benchmark. This is not a reason to lower expectations, but is important to acknowledge this nuance.
- minor comment, on lines 126-127 where the number of units per country exceeding 1 DALY per year, it would be helpful to include the denominators.

Reviewer #5 (Remarks to the Author):

In general this is a well put together paper which provides some important insights, especially as relates the distribution of burden of LRI across sub-Saharan Africa.

Previous reviews of the manuscript provided substantial comment and revision which resulted in a much improved manuscript and also resolved many minor issues with clarity and content. As such I have no major issues with the manuscript. The major claim of the paper is to be the first paper to resolve a joint analysis of the burdens of LRI, diarrhea, and malaria at the admin 2 level for SSA. This seems an accurate claim. The findings are novel and of interest to the community. The work is convincing and it is well supported by the analysis documented herein. The paper will potentially influence thinking especially in the importance of LRI intervention in SSA.

I have only minor issues to raise with the paper, one is that it as a manuscript is slightly hard to follow, in the sense that it would be easy to come away from the manuscript reading missing some of the main points that the authors convey extremely clearly in the rebuttal letter included in this package. Namely this refers to the findings around the differences in spatial distribution of these three causes. Aligned with this comment is a notion that the figures, which are elegant, require substantial effort to interpret.

I do not have constructive comments on these notes for the authors unfortunately, as conveying the info here is complex and difficult, but I do think that the paper might benefit from a look at restructuring / rewriting some sections for impact on the expected audience.

A couple of minor issues of note:

- 1) I was unable to find the 'code' for this analysis at the link provided. It would be suitable to provide a more direct link to the specific code used for this analysis, rather than the entirety of the GBD study, since readers here might be interested in building upon or validating this work.
- 2) I also struggled with the idea of the counterfactual as moving everything to lowest burden district. This seems massively over optimistic to me in a way that is likely to rub readers the wrong way. Eg. bringing the malaria burden of Kenyan districts all to levels consistent with districts with altitudes too high to sustain malaria transmission seems to be only consistent with full elimination of disease, unlikely for LRI and diarrhea even more so than for malaria. This would require changes to factors that are not intervenable. The authors could likely at minimum make this fairly extreme limitation/choice more apparent in the discussion of these results.

Reviewer #3 (Remarks to the Author):

I believe the paper requires a major rewrite in terms of both the scientific finding and analysis methods, in order for it to be suitable for publication. I thank the authors to respond to my previous comments. However, I find the response to be usually not sufficient. I do agree the efforts made by the authors are admirable, and this paper may have an audience somewhere, but the goals of the paper is beyond what analysis and evidence in this paper could possibly support.

1. In the response, the authors argue the primary aim for the paper "was to highlight these sorts of differences (the different split of burdens spatially)" and the "counterfactual analysis is one approach to highlight the resulting estimates". However, I do think to most readers, the current form of the paper places very strong emphasis on the counterfactual analysis (and it is the only 'new analysis' as well). This is troubling to me as while the authors acknowledge the counterfactual target is 'optimistic', I would argue they are 'misleading'. Finding the best Admin-2 area in a country is a lazy target that means different things in different country: whether it is achievable goal? Are you basing the counterfactual scenario on a capitol urban administrative unit that is largely different from the entire rest of the country? Or are you basing the scenario on a lowly populated unit with highly uncertain estimates? The point here, is not whether the target is optimistic, but rather the choice of the counterfactual is heterogeneous and uninterpretable when comparing across countries. So by presenting the approach and estimates for sub-Saharan Africa as a whole, I think it is misleading to the audience.

One of the important findings in subnational disease burden estimation is that there are large inequalities within countries. The counterfactual analysis that is part of our overall body of research presented in this paper is meant to highlight those while also providing specific numbers of excess disability-adjusted life-years (DALYs) that may have been averted if rates of disease were shared across all subnational units within a country. We have considered the Reviewer's feedback that comparing all subnational units to the best performing within a country may be unrealistic for a number of reasons, including some that the Reviewer provided.

In this Revision of the manuscript, we have compared the observed rates of disease with the median rates within countries. This process is completed at the draw level to maintain uncertainty in our findings. Using the median disease rates as our counterfactual could be considered a baseline. The fact that 50% of subnational units do better than that is, in our opinion, a strong case that the remaining 50% could do better at preventing disease burden. The result is that many millions of children, in our estimates, have unnecessarily experienced disease burden due to diarrhea, LRIs, and malaria in sub-Saharan Africa.

2. While the previous comment might be addressed by country-specific analysis, which is not the purpose of this paper, but could be done if someone put in the effort. A bigger concern I have is the validity of the general methodology. The authors in the response argued that the multiple spatial surface are substantially different, and thus are suitable for the analysis where they treat the three sets of estimates as independent. I am not convinced the authors fully understand my concern. Dependence and 'how similar the posterior means are on the map' measures two different things. Any covariates and input shared in the three models would lead to dependences in the estimates, as any uncertainties in those covariates would propagate through to all the final estimates. If there are a large number of overlapping covariates used, the estimates will be highly correlated. In addition, the fact that spatial and temporal smoothing models are used also lead to common unmeasured confounders being captured by the random effects in the three models. Both could be very difficult to quantify after fitting three

models separately. Thus it seems to me that treating the three analysis as independent could potentially both mess up the uncertainty of the final joint analysis, and lead to bias in the conclusion. Finally, while as I emphasized already, looking at Figure 1 cannot prove/illustrate independence, I want to highlight the similarities of the Diarrhea and LRI map, which I would not think as 'quite different', and certainly would consider the chance of it being driven by modeling (in addition to the true co-occurrence) to be non-ignorable.

We strongly disagree with the suggestion that this analysis presented in this publication was “lazy” or could be improved if someone “put in the effort”. The suggestions the Reviewer provides are not the intended purpose of this analysis. We agree that an analysis of subnational variation at the country level could yield important findings for policy and improving child health. In fact, we have published findings exactly like that. Because of that experience, we know that the requirements for such a publication are different and well beyond the scope of this paper. The scope and purpose of this analysis is to provide descriptive estimates of overlapping disease burden across the entire sub-Saharan African region; to illustrate where diarrhea, LRI, and malaria disease burden overlap and where they don't; and to challenge public health stakeholders to focus on subnational inequalities and variation in disease burden.

The Reviewer's concerns about independence are noted, especially in regard to the shared covariates. While we completely agree that there is a difference between “dependence” and “how similar the posterior means are on the map”, certainly a high correlation between geospatial estimates would be a symptom of the issues the reviewer raises. The reviewer's comment is slightly confusing as directly after saying correlation in outcomes is different from dependence, the reviewer argues “If there are a large number of overlapping covariates used, the estimates will be highly correlated”. The demonstrable fact that the outcomes are not highly correlated (as that is one of the main points of the paper) again supports the claim that treating the final three estimates independently is not overly biasing.”

In our view, there are several layers to dependence between the disease burden:

1. Shared covariates in geospatial models. It is true that some covariates are shared between the models including environmental, built environment, maternal education, and others. Unmeasured confounding is always a concern in epidemiological analyses but we are not in agreement that our results would be biased by how this variation is captured in the random effects of each disease model.
2. Enforcing consistency in Global Burden of Disease estimates at the national level. The sum of disease burdens in our analysis must match the national-level estimates from the GBD study. The Reviewer may be familiar with the GBD assumption that deaths may only be attributed to one underlying cause. This means that the sum of cause-specific deaths must match the all death total in a given country-year. Because the sum of the subnational estimates match at the national level, we have not counted years of life lost (YLLs) multiple times. There is no such assumption about years lived with disability (YLDs) resulting from incident infection in the GBD (but overlap may be plausible; detailed in point 4 below).
3. Availability of data. Some countries, like the Central African Republic, have very little survey or study data and are more highly dependent on covariates and GBD estimates. We are fairly confident from other sources- such as total under-5 mortality and Human Development Indicators- that DALYs in CAR are among the highest in the world. This almost certainly contributes to the high burden of both diarrhea and LRI.

4. **Biological and political plausibility in overlapping disease burden. There is a body of evidence that infectious diseases make children more vulnerable to other infections, either through chronic inflammation or growth faltering. Subnational areas within countries that have been neglected, are in conflict, or generally resource poor are more likely to have larger disease burden across the three causes.**

Finally, fitting the joint distribution across the three 3-dimensional surfaces simultaneously is well beyond the scope of the current work. Such an endeavor is of interest, but there is considerable model development required before such an analysis could be conducted. A short reflection on the value of such methods is included on page 14, lines 310-315.

Reviewer #4 (Remarks to the Author):

This research calculates DALYs for 3 or the major causes of under 5 morbidity and mortality across 43 sub-Saharan African countries to the second administrative unit to understand their co-occurrence. The finding that there is a substantial amount of heterogeneity in disease patterns across and between countries supports the authors conclusions of the need for a more tailored approach to maximise the potential for impact in reducing disease burden. This work will be influential in the growing recognition of the potential for integrated disease surveillance and control. The authors have adequately responded to the comments from the previous review and I only have a couple of additional comments for consideration.

- The disease-specific models applied for this analysis have been previously published and methods used have been adequately summarised here. However, the models are ultimately reliant on the availability and quality of data on each of the three diseases being assessed. Although the issue of data quality is briefly acknowledged in the limitations paragraph, it would be helpful to present the results in a way where the data quality issues over time and between countries are better acknowledged.

Although the intention with this publication is about a synthesis of previously reported models, we recognize that evaluation of the models themselves may play an important role in understanding the findings.

Please find information about the data coverage in each individual model here:

- **Malaria (page 26-54)**
- **Lower respiratory infections (page 45-51)**
- **Diarrhea (pages 37-42)**

- Shifting the focus from vertical programs to a more integrated and data-driven perspective on reducing disease-burden is an important finding. However, for programs to inform interventions tailored to the distribution of multiple diseases, the data that is typically available is case/death counts per unit and not DALYs or modelled estimates that can smooth over data quality issues or combining active and passive surveillance data as per the malaria analysis presented. How could the observed trends translate into a more operational approach accessible to programs? This is touched upon in lines 300-304 but could be expanded from a more operational perspective (e.g. could programs simply map each disease counts independently and compare the patterns?).

Thanks for this helpful comment. We chose to compare these diseases using disability-adjusted life-years (DALYs) because they are directly comparable between not just other infectious diseases but also against neonatal diseases, injuries, non-communicable diseases, and other sources of health loss. Given that the diseases assessed in this analysis tend to have a relatively short duration of illness and because we are focused on children younger than 5 years, nearly all the DALYs are from years of life lost (YLLs) which should be closely comparable to death counts.

Due to the dearth of reliable mortality and cause of death data across much of sub-Saharan Africa, we have not systematically estimated case-fatality ratios or other similar metrics. This type of data would be extremely informative both for modeling and for operations and programmatic decisions regarding public health resources and interventions. Stronger surveillance systems would reduce the amount of modeling that analyses like ours need to conduct.

- the decision for selecting the benchmark for comparison and the limitations are well explained (as per the previous reviewers comments). However, I think one point that doesn't come out very well is the heterogeneity of disease burden in the country impacting how easy it is to achieve the national benchmark. This is not a reason to lower expectations, but is important to acknowledge this nuance.

Thanks for this feedback. We have modified the analysis to benchmark against each country's median disease burden. As noted in the response to Reviewer 3, comparing against the median might be more useful than against the best performing subnational unit.

- minor comment, on lines 126-127 where the number of units per country exceeding 1 DALY per year, it would be helpful to include the denominators.

Great suggestion! We also added the percentage of all units.

Reviewer #5 (Remarks to the Author):

In general this is a well put together paper which provides some important insights, especially as relates the distribution of burden of LRI across sub-Saharan Africa.

Previous reviews of the manuscript provided substantial comment and revision which resulted in a much improved manuscript and also resolved many minor issues with clarity and content. As such I have no major issues with the manuscript. The major claim of the paper is to be the first paper to resolve a joint analysis of the burdens of LRI, diarrhea, and malaria at the admin 2 level for SSA. This seems an accurate claim. The findings are novel and of interest to the community. The work is convincing and it is well supported by the analysis documented herein. The paper will potentially influence thinking especially in the importance of LRI intervention in SSA.

I have only minor issues to raise with the paper, one is that it as a manuscript is slightly hard to follow, in the sense that it would be easy to come away from the manuscript reading missing some of the main points that the authors convey extremely clearly in the rebuttal letter included in this package. Namely this refers to the findings around the differences in spatial distribution of these three causes. Aligned with this comment is a notion that the figures, which are elegant, require substantial effort to interpret.

We have made revisions to many sections of the manuscript with an eye on being more precise with our language and to add context in some areas that may have lacked it in earlier revisions. One concrete example is creating sub-sections within the model results section to break apart the themes we discuss in each.

Creating figures is at least partly an art. We have tried to be thoughtful in balancing the large amount of information contained in each with simple and intuitive to understand plots. It seems we haven't succeeded as much as we hoped! If there are specific suggestions or recommendations, we would be eager to revisit the figures. Otherwise we anticipate some changes if the manuscript is accepted and the graphical illustrators begin working on adapting to the Journal's style.

I do not have constructive comments on these notes for the authors unfortunately, as conveying the info here is complex and difficult, but I do think that the paper might benefit from a look at restructuring / rewriting some sections for impact on the expected audience.

A couple of minor issues of note:

1) I was unable to find the 'code' for this analysis at the link provided. It would be suitable to provide a more direct link to the specific code used for this analysis, rather than the entirety of the GBD study, since readers here might be interested in building upon or validating this work.

The code for the geospatial models for each disease are available here. The code for the counterfactual analysis in this paper will be made available upon acceptance/publication. If desired, we can share the code to conduct the counterfactual analysis directly with the Reviewer.

2) I also struggled with the idea of the counterfactual as moving everything to lowest burden district. This seems massively over optimistic to me in a way that is likely to rub readers the wrong way. Eg. bringing the malaria burden of Kenyan districts all to levels consistent with districts with altitudes too high to sustain malaria transmission seems to be only consistent with full elimination of disease, unlikely for LRI and diarrhea even more so than for malaria. This would require changes to factors that are not intervenable. The authors could likely at minimum make this fairly extreme limitation/choice more apparent in the discussion of these results.

Thanks for this feedback, it is common across the Reviewers! We have adjusted our counterfactual to compare against the median disease burdens within each country. Although we still believe that policy makers and public health stakeholders should aggressively strive to reduce disease burden and disease burden inequalities across all subnational units, comparing to the best performing subnational unit may not be representative of the entire country or even possible given resource and other constraints. We have presented our analysis of a counterfactual to the median as a bare minimum expectation, that reducing disease burden to that level should be achievable and is urgent to meet global goals in under-5 mortality.

REVIEWERS' COMMENTS

Reviewer #3 (Remarks to the Author):

I thank the authors for addressing the first major comment about benchmarking target. I do recognize the efforts needed to address the second comment might be substantial and beyond the intended scope of the work, while I still remain skeptical about the authors' approach. The authors did not respond to my query about the similarity between IRL and Diarrhea map and argues that outcomes are not highly correlated. In my opinion, the authors acknowledged the sources of dependence that could make the estimates provided in this manuscript biased, but did not try to (at least partially) assess their impacts. I remain unconvinced about the authors' argument that the proposed study is sufficient for the purpose of `` provide descriptive estimates of overlapping disease burden across the entire sub-Saharan African region; to illustrate where diarrhea, LRI, and malaria disease burden overlap and where they don't; and to challenge public health stakeholders to focus on subnational inequalities and variation in disease burden."

Reviewer #4 (Remarks to the Author):

Thank you for considering and addressing my comments in the revised manuscript. I have only 1 minor additional comment to consider. On lines 104-105, the authors state the aim of this work is around the "...potential impact of reducing subnational heterogeneity among these three causes on childhood survival". Is the aim of reducing heterogeneity within countries an important goal (i.e. if everyone increased and met at the maximum, the heterogeneity would be reduced but the impact on public health would be detrimental. Alternatively, should the emphasis be on reducing burden for each disease in the most efficient and effective way possible? This would involve some minor tweaks to the text in a couple of places but is a pretty major conceptual difference.

Reviewer #5 (Remarks to the Author):

This revision of the manuscript is sound, well written clearer and much more defensible in the sensitivity analysis than the previous version. I have no additional substantive comments.

Reviewer #3 (Remarks to the Author):

I thank the authors for addressing the first major comment about benchmarking target. I do recognize the efforts needed to address the second comment might be substantial and beyond the intended scope of the work, while I still remain skeptical about the authors' approach. The authors did not respond to my query about the similarity between IRL and Diarrhea map and argues that outcomes are not highly correlated. In my opinion, the authors acknowledged the sources of dependence that could make the estimates provided in this manuscript biased, but did not try to (at least partially) assess their impacts. I remain unconvinced about the authors' argument that the proposed study is sufficient for the purpose of "provide descriptive estimates of overlapping disease burden across the entire sub-Saharan African region; to illustrate where diarrhea, LRI, and malaria disease burden overlap and where they don't; and to challenge public health stakeholders to focus on subnational inequalities and variation in disease burden."

While we do not agree with the reviewer that there is considerable similarity between the LRI and Diarrhea maps (their dissimilarity is one of the main points of the paper), we do acknowledge that given the overlap in covariates of each model, there are some reasons for concern in theory and a deep dive into the relative importance of each covariate for each model could be performed. As such, we have added the following sentence on lines 316-318:

From a methodological standpoint regarding the simultaneous modelling of multiple cases, care must be taken to balance model performance with inferentiality as many causes share the same underlying drivers.

Reviewer #4 (Remarks to the Author):

Thank you for considering and addressing my comments in the revised manuscript. I have only 1 minor additional comment to consider. On lines 104-105, the authors state the aim of this work is around the "...potential impact of reducing subnational heterogeneity among these three causes on childhood survival". Is the aim of reducing heterogeneity within countries an important goal (i.e. if everyone increased and met at the maximum, the heterogeneity would be reduced but the impact on public health would be detrimental. Alternatively, should the emphasis be on reducing burden for each disease in the most efficient and effective way possible? This would involve some minor tweaks to the text in a couple of places but is a pretty major conceptual difference.

The reviewer's comment is well taken and we have altered the sentence as follows:

By using each country's median burden in each year as a local (nationally specific) benchmark, we show for the first time the potential impact of reducing subnational heterogeneity by improving health outcomes for those most vulnerable among these three causes on childhood survival